# A discovery platform for identification of host-induced bacterial biosensors from diverse sources

Clare M Robinson[1,2,4], David Carreño[1,2], Tim Weber[1,5], Yangyumeng Chen[1,6] & David T Riglar [ID] [1,2,3 ✉]

## Abstract

**Synthetic biology approaches such as whole-cell biosensing and 'sense-and-respond' therapeutics aim to enlist the vast sensing repertoire of gut microbes to drive cutting-edge clinical and research applications. However, well-characterised circuit components that sense health- and disease-relevant conditions within the gut remain limited. Here, we extend the flexibility and power of a biosensor screening platform using bacterial memory circuits. We construct libraries of sensory components sourced from diverse gut bacteria using a bespoke two-component system identification and cloning pipeline. Tagging unique strains using a hypervariable DNA barcode enables parallel tracking of thousands of unique clones, corresponding to ~150 putative biosensors, in a single experiment. Evaluating sensor activity and performance heterogeneity across various in vitro and in vivo conditions using mouse models, we identify several biosensors of interest. Validated hits include biosensors with relevance for autonomous control of synthetic functions within the mammalian gut and for non-invasive monitoring of inflammatory disease using faecal sampling. This approach will promote rapid biosensor engineering to advance the development of synthetic biology tools for deployment within complex environments.**

**Keywords** Bacterial Biosensor; Gut Microbiome; Synthetic Biology; Inflammation
**Subject Categories** Biotechnology & Synthetic Biology; Microbiology, Virology & Host Pathogen Interaction

## Introduction

A longstanding vision for synthetic biology is the development of bacterial whole-cell biosensors and 'sense-and-respond' live biotherapeutic products. These systems promise to autonomously sense and induce reporters or therapeutics under defined spatial, dietary and disease conditions (Riglar and Silver, 2018; Robinson et al, 2022). Potential applications include diagnostics, disease monitoring and biotherapeutics (Riglar and Silver, 2018). However, design,

construction and testing of new bacterial biosensors is non-trivial. Sensors that have been prototyped and function well in controlled laboratory settings often perform poorly when exposed to the challenging growth and metabolic conditions of complex 'real-world' environments, such as the mammalian gut. Individual testing of new biosensors in vivo is also a resource-intensive process, incurring high costs in the form of money, time, and number of animals used in research. Consequently, well-characterised biosensing circuits for the mammalian gut remain limited, hindering the complete fulfilment of this vision (Barra et al, 2020; Tanna et al, 2021).

Direct measurement of metabolic and transcriptional biomarkers in the gut is also challenging. The brevity of bacterial mRNA half-lives, which can average less than a minute (Jenniches et al, 2024), considerably limits the insights that can be gained from faecal bacterial transcriptional sequencing. Similarly, rapid spatio-temporal dynamics of the gut microbiota and microbiota-linked metabolites likely make the quantities of many metabolites in faeces unrepresentative of internal gut conditions (preprint: Carreño et al, 2024). Synthetic genetic memory circuits, which convert transient signals into sustained responses, for example through transcriptional switches (Kotula et al, 2014; Riglar et al, 2017) or DNA editing (Inda-Webb et al, 2023; Mimee et al, 2015; Schmidt et al, 2022; Zou et al, 2023), are powerful tools for mitigating these challenges and providing non-invasive biosensor discovery, reporting and actuation. In rodent models, memory circuits have been used to successfully track inflammatory signals (Naydich et al, 2019; Riglar et al, 2017; Schmidt et al, 2022; Zou et al, 2023), record microbial response to dietary perturbations (Mimee et al, 2015; Schmidt et al, 2022), detect tumour DNA (Cooper et al, 2023), and actuate sustained biotherapeutic secretion (Zou et al, 2023).

High-throughput in vivo screening addresses both the failure rates of in vitro prototyped sensors and reduces the resources required to test sensors under in vivo conditions. We previously developed a high-throughput memory system (HTMS) in mouse commensal *Escherichia coli* NGF-1 (Ziesack et al, 2018) to screen *E. coli* promoters as biosensor candidates within the mouse gut (Naydich et al, 2019). Memory recording is based on λ-phage's lysis-lysogeny switch (Kotula et al, 2014). In the chassis' memory 'OFF' state, *cI* represses β-galactosidase (*lacZ*) and streptomycin/spectinomycin adenyltransferase (*aadA*) reporter cassettes (Fig. 1A, left). Upon exposure to an inducing signal, the sensing component triggers expression of cI dominant negative (cI[DN]), switching cells into memory 'ON' states, in

[1]Department of Infectious Disease, Imperial College London, London SW72AZ, UK. [2]The Francis Crick Institute, London NW11AT, UK. [3]Imperial Centre for Engineering Biology, Imperial College London, SW72AZ London, UK. [4]Present address: Full Circle Labs Ltd, Venture X White City, London W127SL, UK. [5]Present address: Department of Molecular Life Sciences, University of Zurich, Zurich CH-8057, Switzerland. [6]Present address: Max Planck Institute for Heart and Lung Research, Bad Nauheim 61231, Germany.
✉E-mail: d.riglar@imperial.ac.uk

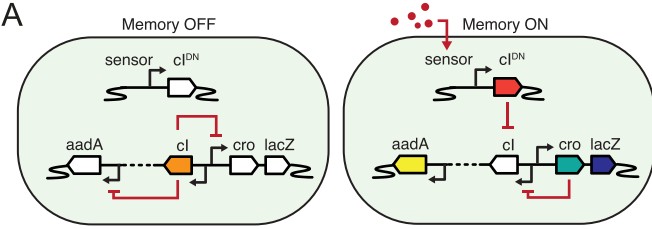

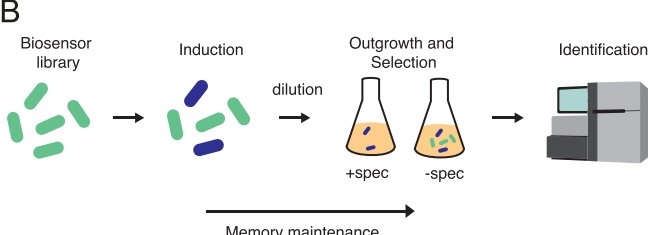

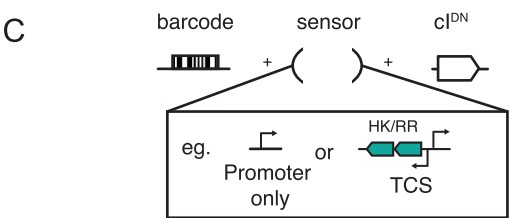

**Figure 1. A high-throughput transcriptional screening pipeline based on an engineered memory circuit to identify novel inducible circuits.**

(A) A λ-phage-based memory circuit. In the presence of an activating stimulus, $cI^{DN}$ is expressed and inhibits $cI$, de-repressing $cro$ and allowing expression of reporter genes $lacZ$ and $aadA$. Cro expression maintains this memory ON state irrespective of ongoing exposure to the activating stimulus. (B) A library of memory-linked bacterial biosensors can be rapidly screened in parallel for activated sensors by growth in spectinomycin selective media followed by massive parallel sequencing of recovered gDNA for identification. (C) Various sensing components of differing lengths and complexities can be assembled in parallel to drive $cI^{DN}$ expression. Tagging of each construct with a unique, hypervariable, DNA barcode facilitates efficient sensor identification and library flexibility.

capable of detecting various molecular signals (Shaw et al, 2022). They are typically composed of a membrane-bound histidine kinase (HK) and a partner response regulator (RR) that controls target gene transcription (Jacob-Dubuisson et al, 2018). TCSs are of particular interest as engineered sensing components because novel capacities can be transferred between bacterial species through simple introduction of TCS genes (Daeffler et al, 2017; Riglar et al, 2017). Outside of *E. coli* grown in laboratory conditions, the vast majority of TCSs and their regulons remain poorly characterised, representing a large untapped pool of novel sensing capabilities. On a technical level, TCSs also provide a pool of sensor components with varying size that are challenging to track within a library context.

DNA barcodes can be used to effectively identify clones within a library via short-read massive parallel sequencing. The approach has several advantages, including (i) cost-effective, low-bias, simultaneous tracking of sensors, with broad flexibility for sensor size and genetic format; (ii) the ability to reliably and accurately identify sensors with highly similar sequence composition, for example promoters with ribosome binding site (RBS) variations that may differ by only a few base pairs; (iii) increased confidence in sensor response data through construction of multiple clones with identical sensor sequence but unique barcodes for measurement in a single animal or test condition; and (iv) increased flexibility during testing for adding new sensors to existing libraries and/or testing combined libraries in a single experiment.

Here, we build two new libraries of sensor components: the first focused largely on TCSs from diverse bacterial origins (Library 1) and a second predominantly containing *E. coli*-derived sensors targeted to be responsive to the mammalian gut (Library 2). During multiplexed DNA assembly of sensors to drive $cI^{DN}$ expression, each construct is tagged with a unique hypervariable DNA barcode, which can be used for subsequent clone identification (Fig. 1C). We test these libraries in vitro and in vivo in the murine gut, identifying and validating several new sensors of interest. Included are sensors responsive to the gut environment both generally and specifically during inflammation. Together, this work advances methodology for the application of library-based screens of synthetic biology components in the context of a conventional mouse gut microbiota. Our results demonstrate the power of this approach for identifying context-dependent sensors that can actuate synthetic circuits under specific conditions within the mammalian gut.

## Results

### Construction of bacterial biosensor libraries encoding diverse native and heterologous sensing machinery

Most HKs are encoded within 200 bp of their partner RR (Williams and Whitworth, 2010). Where a HK-RR pair sit next to a diverging gene, we term this a 'grouped' TCS. Many grouped TCSs regulate their adjacent and divergent genes. Of a total of 30 known TCS pairs in *E. coli* MG1655, 13 lie in this 'grouped' formation, and of these, 10/13 regulate the divergent, adjacent gene based on Ecocyc annotations (Keseler et al, 2021). We, therefore, focused on these 'grouped' TCSs for biosensor library expansion due to the ability to replicate complete sensing machinery (HK, RR and putatively regulated promoter sequence) in a single PCR from target strain genomic DNA for transfer to a heterologous host.

which the repression of *cro*, *lacZ* and *aadA* is relieved (Fig. 1A, right). Repression of *cI* by cro maintains switching status irrespective of the presence of the induction signal, resulting in cellular 'memory'. Memory 'ON' bacteria can thus be selected via growth in the presence of spectinomycin antibiotics or detected via β-galactosidase enzymatic activity. The screening process involves (i) induction of pooled libraries of HTMS bacteria, each with memory driven by a different sensor; followed by (ii) dilution into two parallel in vitro cultures, one of which undergoes spectinomycin selection to isolate memory 'ON' bacteria; and (iii) identification of sensors in each culture via massive parallel sequencing (Fig. 1B).

We now expand the power and flexibility of this screening platform to accommodate a diversity of sensory components from varied microbial sources. These include, but are not limited to, multi-gene heterologous two-component systems (TCSs), for which we develop a custom computational pipeline for rapid identification of cloning-compatible systems. Bacterial TCSs are modular signalling pathways

**Table 1. Compatible TCSs computationally identified and cloned from diverse gut bacteria.**

| Strain | Total TCSs | 'Grouped' | SapI-compatible | BsaI compatible (not SapI) | Successfully amplified | Assigned Library 1 | Assigned Library 2 |
|---|---|---|---|---|---|---|---|
| *Bacteroides caccae* | 9 | 5 | 5 | 0 | 4 | 1 | 1 |
| *Bacteroides dorei* | 14 | 7 | 3 | 4 | 2 | 0 | 2 |
| *Bacteroides fragilis* | 13 | 8 | 6 | 0 | 5 | 2 | 0 |
| *Bacteroides thetaiotaomicron* | 12 | 8 | 3 | 3 | 1 | 1 | 0 |
| *Blautia producta* | 90 | 26 | 19 | 6 | 14 | 5 | 5 |
| *Citrobacter rodentium* | 22 | 14 | 11 | 2 | 13 | 11 | 2 |
| *Clostridium difficile* | 43 | 8 | 8 | 1 | 3 | 0 | 1 |
| *Edwardsiella tarda* | 20 | 11 | 8 | 1 | 5 | 5 | 0 |
| *Enterococcus faecalis* | 13 | 4 | 3 | 0 | 3 | 2 | 0 |
| *Klebsiella oxytoca* | 25 | 12 | 8 | 5 | 2 | 1 | 0 |
| *Klebsiella pneumoniae* | 25 | 12 | 10 | 0 | 8 | 7 | 0 |
| *Lactobacillus gasseri* | 6 | 1 | 1 | 0 | 1 | 1 | 0 |
| *Lactobacillus plantarum* | 13 | 2 | 2 | 1 | 2 | 1 | 0 |
| *Ruminococccus gnavus* | 36 | 13 | 8 | 4 | 6 | 2 | 3 |
| *Salmonella* Typhimurium | 22 | 11 | 6 | 2 | 5 | 3 | 0 |
| *Vibrio cholerae* | 24 | 13 | 8 | 3 | 5 | 3 | 2 |
| *Yersinia enterocolitica* | 20 | 8 | 7 | 1 | 5 | 4 | 0 |
| Total | 407 | 163 | 116 | 33 | 84 | 49 | 16 |
| | | | | | | Total TCS | 65 |

To enable rapid library construction, we developed a computational script to extract grouped TCSs from available genomic sequencing data: (i) TCS components were identified from each genome using the Microbial Signal Transduction Database (MiST 3.0) (Gumerov et al, 2020); (ii) HK-RR pairs were then identified by close proximity (<200 bp) on the same DNA strand; and, (iii) Gene IDs of all pairs adjacent to a diverging gene were then extracted from NCBI, along with genomic sequence data for the entire grouped sensor region. Using this pipeline directed against 17 bacterial species for which we had access to both full genome sequences and gDNA of exact, or closely related, strains to use as PCR templates, we identified 163 'grouped TCSs' (Table 1).

Identified TCSs were then cloned so that the divergent promoter would drive cI$^{DN}$ expression and thus trigger our λ-memory system when activated (Fig. EV1A). Construction of large libraries of individually cloned barcoded sensors is unfeasible due to cost and time constraints. Therefore, sensors were constructed by pooled golden gate assembly which uses Type IIS restriction enzymes, such as SapI and BsaI, that cut outside their recognition sites to afford ordered and scarless assembly of multiple DNA fragments in a single pot (Engler et al, 2008) (Fig. EV1A). In total, 146/163 identified TCSs were compatible with a SapI-based golden gate cloning strategy (Table 1). These sensors were then amplified from gDNA by PCR and pooled for cloning. To seamlessly incorporate unique barcodes into each plasmid generated during pooled cloning, we synthesised a 106 bp randomised nucleotide hypervariable barcode, flanked by sequencing adapter sites and Golden Gate cloning sites. This sequence was ordered as a single ultramer oligonucleotide and duplexed via a short-cycle PCR reaction for compatibility with Golden Gate assembly. Once assembled, the

sensor library was genome-integrated into *E. coli* NGF-1 encoding the high-throughput λ-memory system (PAS811) via Tn7 insertion (Naydich et al, 2019) (Fig. EV1A).

High-fidelity sequences of the barcodes present in each library were identified by clustering of massive parallel short-read sequencing (Fig. EV1B). These barcodes were linked with their associated sensing components through massive parallel long-read sequencing, which spanned adjacent sensor and barcode regions (Fig. EV1B). Barcodes were assigned for 49 unique SapI-compatible TCSs. We supplemented these strains with 9 additional individually cloned 'curated' sensors identified in our previous studies (Naydich et al, 2019; Riglar et al, 2017) to afford 'Library 1' (Table 1; Dataset EV1). Included were (i) ST TCS1*: a *Salmonella enterica* subsp. *enterica* serovar Typhimurium (herein *S.* Typhimurium) *ttrSR*-P$_{ttrBCA}$ sensor, edited with a synthetic ribosome binding site, that followed the grouped TCS conformation and previously showed response to the inflammatory biomarker tetrathionate during murine inflammation (Riglar et al, 2017) and (ii) Ec ynfE15: the *E. coli* Nissle 1917 P$_{ynfEFGH}$ promoter edited with a synthetic 5' untranslated region and MCD15 ribosome binding site (Mutalik et al, 2013), that previously showed switching in the healthy murine gut (Naydich et al, 2019). Together, these 58 sensors making up Library 1 were linked to 504 assigned unique barcodes.

Building on the flexibility of this method to easily combine sensors, barcodes and libraries, we subsequently cloned Library 2, consisting of: (1) *E. coli* promoter regions previously induced in mouse models, including during inflammation in monocolonised mice (Schmidt et al, 2022); (2) computationally identified TCSs that were not successfully assigned in Library 1 construction, including those with BsaI but not SapI cloning compatibility; and (3) several

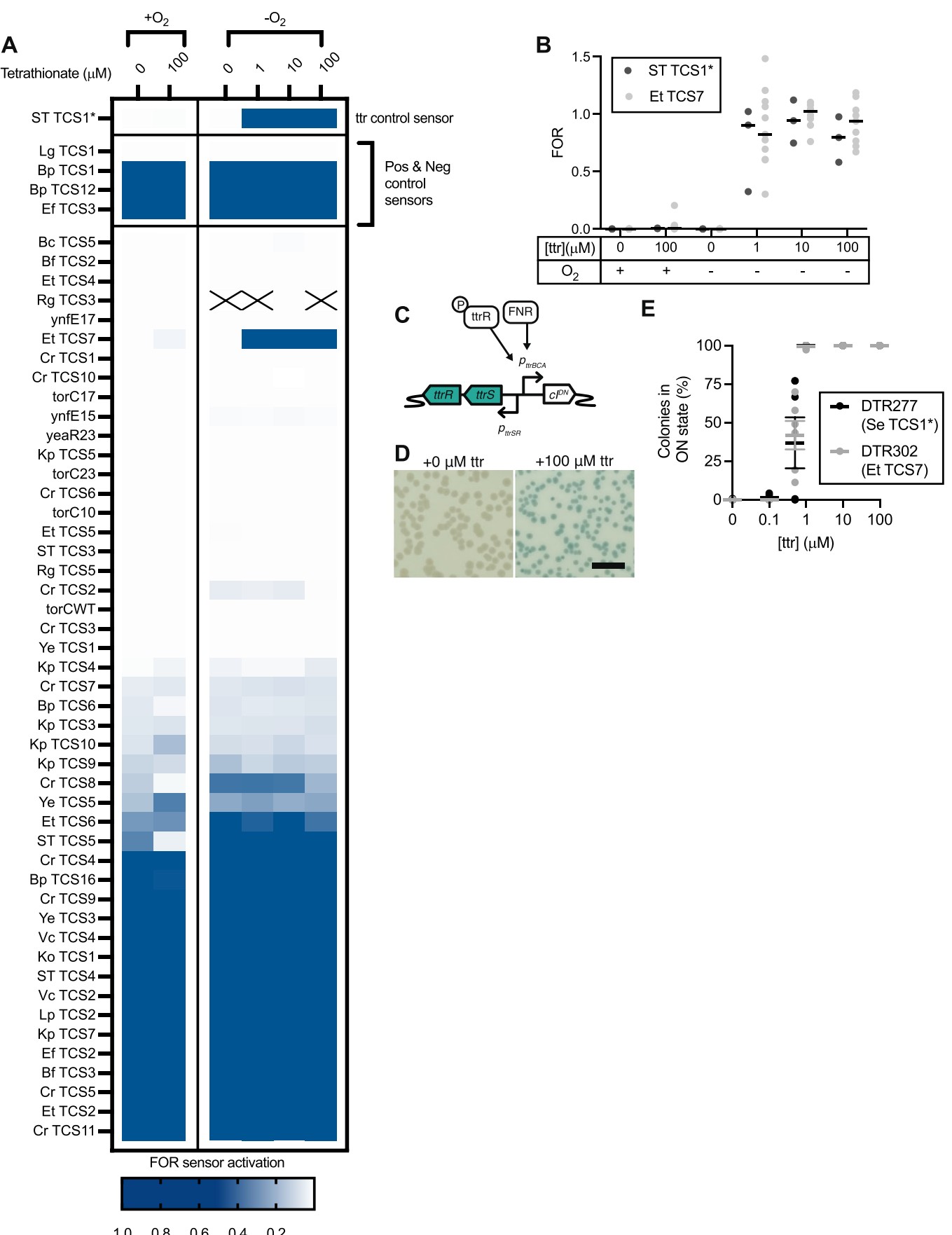

◄ **Figure 2.  Barcoded biosensor screening of Library 1 identifies specific biosensor responses in vitro.**

(A) Heatmap of median fractional odds ratio (FOR) for Library 1 sensors screened during in vitro growth in the presence and absence of oxygen and varying levels of tetrathionate (ttr). Squares marked X indicate insufficient sequencing read data for that sensor and condition. (B) Activation (FOR) of individual barcodes linked to ST TCS1* (n = 3 barcodes) and Et TCS7 (n = 9–11 barcodes per timepoint) sensors under the same varying oxygen and tetrathionate growth conditions. Medians of all barcodes are marked. (C) Putative sensor regulation of both Et TCS7 and ST TCS1*. (D) DTR302, an individually cloned Et TCS7 sensor strain, demonstrated no evidence of switching in the absence of tetrathionate and 100% switching following anaerobic growth in 100 mM tetrathionate, as measured by indicator plating. Scale bar = 0.5 cm. (E) Response dynamics of DTR302 and DTR277 (an individually cloned ST TCS1* sensor strain) were similar across growth in various tetrathionate concentrations as measured by indicator plating. The graph shows counts from 6 biological replicates of each sensor with mean ± SEM overlaid. Source data are available online for this figure.

additional curated sensors of interest (Table 1; Dataset EV1). The 17 TCSs and 72 *E. coli* promoter regions of interest in Library 2 were linked to 10,063 assigned unique barcodes.

The number of assigned barcodes per sensor ranged between 1 and 44 (median 5) for Library 1 and 1 and 1001 (median 52) for Library 2 (Fig. EV1C). Each library demonstrated similar abundance profiles (Fig. EV1D). Between the two libraries, we confirmed at least 1 barcode for ~80% of successfully PCR-amplified TCS sensor regions with this method (Table 1).

## In vitro validation confirms barcoded library biosensor screening capabilities

To quantify the state of each sensor, we compared sequencing read counts for each barcode from samples split and grown in the presence or absence of spectinomycin (Fig. 1B). Only ON sensors grow with spectinomycin due to *aadA* expression. The degree of switching is represented as a fractional odds ratio (FOR) by normalisation to reads from known positive and negative strains within the library. FOR ~1 corresponds to a fully ON sensor and ~0 to a fully OFF sensor, with intermediate values corresponding to sensors activated in at least a subset of the population. Incomplete activation may occur when exposure to inducers is limited in concentration or time, due to a promoter having low maximal expression, or from spatially variable regulation across the population, such as may occur within the non-uniform gut. We have previously observed biologically meaningful activation of sensors in as low as ~1% of bacteria within a given population, so will consider FOR >0.01 as potentially activated in our screens.

To validate the barcoded library screening approach, we cultured pooled sensors from Library 1 in vitro with varying tetrathionate concentrations and with or without oxygen (Fig. 2A). As expected, around half of the sensors were consistently ON in all conditions, suggesting a level of baseline expression from these promoters during in vitro growth, resulting in activation of the memory circuit under all conditions. Various sensors showed differential activation between the conditions. Mirroring our previous results, ST TCS1* sensors activated strongly and specifically when grown with tetrathionate in the absence of oxygen but not in the absence of tetrathionate or presence of oxygen (Fig. 2A,B) (Riglar et al, 2017).

## Barcode screening successfully identifies unannotated biosensor candidates

The barcodes linked to *Edwardsiella tarda* TCS7 (Et TCS7) were also consistently induced at levels comparable to ST TCS1* across

the conditions tested (Fig. 2B). While the available gene annotations for the strain used for TCS extraction (*E. tarda* KC-Pc-HB1) did not predict this behaviour, BLAST searches identified homology to a putative tetrathionate reductase promoter region with its associated TCS in *E. tarda* strains Et54 (Data ref: NCBI Protein BAE19899.1 (2005)) and FL6-60 (Daeffler et al, 2017) (Fig. 2C).

All sensors identified as candidates of interest through library-based screens are individually cloned and verified to confirm the specificity and sensitivity of sensor response. ST TCS1* and Et TCS7 sensors were therefore cloned as individual strains (DTR277 and DTR302, respectively) and their response to tetrathionate was confirmed by plating of induced cultures on X-gal indicator plates for blue-white colony screening. When grown anaerobically, both DTR277 and DTR302 demonstrated strong induction in the presence of >1 µM tetrathionate (Fig. 2D,E). This confirms the ability of our screening pipeline to identify unannotated biosensors with specific sensing functions using our screening approach.

## Barcode and strain diversity is maintained over short periods within the murine gut

We have previously demonstrated robust *E. coli* NGF-1 colonisation of the conventional mouse gut, typically following a dose of streptomycin to reduce colonisation resistance, but in select cases also without pre-treatment (Kotula et al, 2014; Riglar et al, 2017). To test colonisation characteristics in our current animal facility, we administered sensor Library 1 by oral gavage (~ $10^{11}$ bacteria/mouse) to groups of 2 healthy, conventional mice 24 h following administration of streptomycin antibiotics (1 mg or 5 mg/mouse as a single dose oral gavage) or mock treatment (Fig. EV2A). Faecal bacterial CFU counts reflected streptomycin provision. Engineered *E. coli* successfully colonised mice provided with 5 mg streptomycin throughout the full 53-day experiment (Fig. EV2B), while engineered bacteria in 1 mg streptomycin or mock-treated animals dropped below the detection limit on day 4–5 and day 3, respectively (Fig. EV2B).

Strain diversity, as estimated by percentage of barcodes identified via sequencing, was high in all animals at day 1 post administration (Fig. EV2C,D). Even in 5 mg streptomycin-treated mice that were colonised effectively, barcode and sensor diversity dropped quickly following administration and by day 4 only 3–4% of total barcoded strains remained (Fig. EV2C,D), with a small number of barcodes dominating colonisation (Fig. EV2E). Consequently, all further library screens focused on short (2-day) experimental timeframes within which barcode diversity was well maintained.

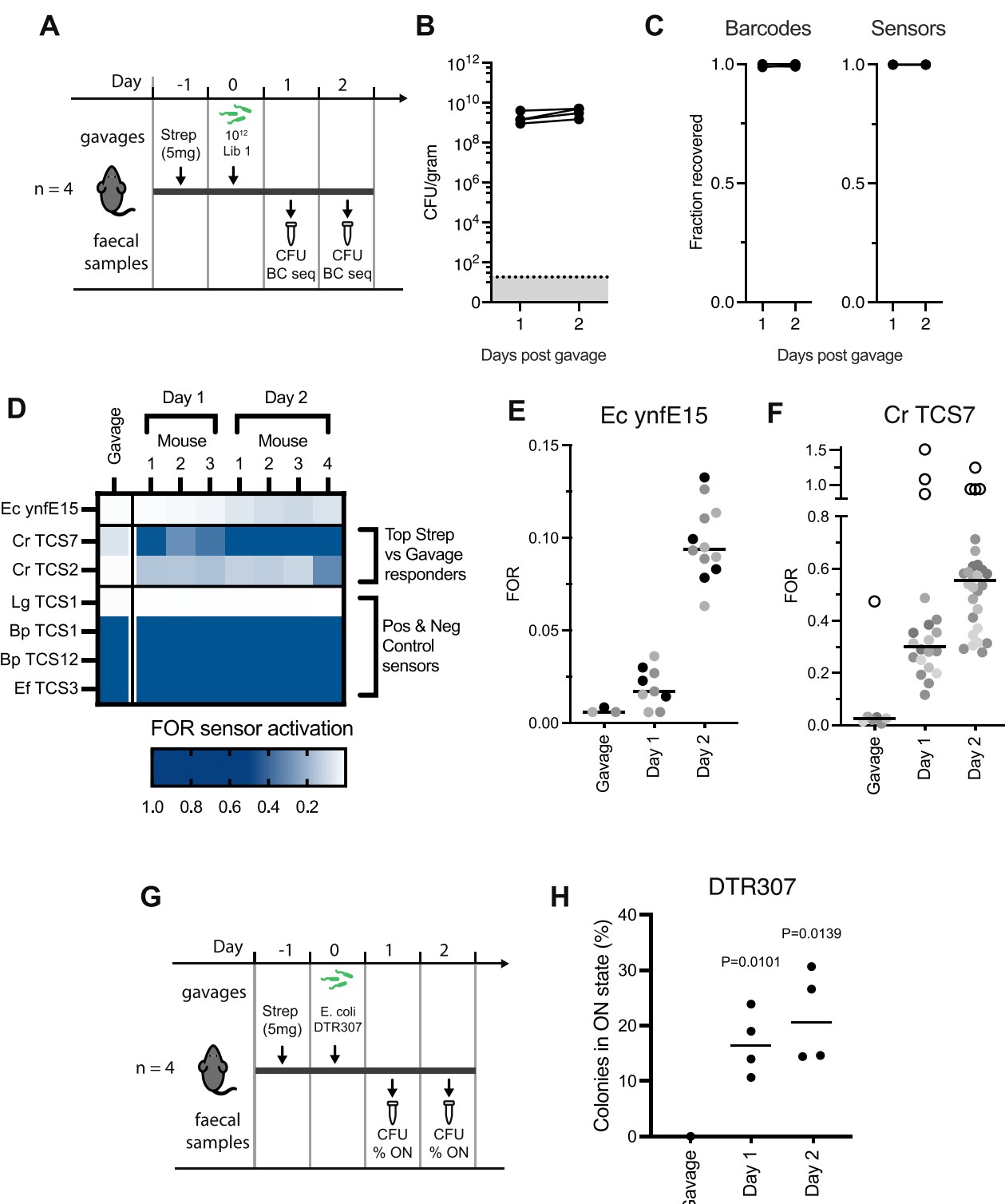

## Barcode screening successfully identifies biosensor response in vivo

To test sensor activation in the gut, we administered Library 1 sensors by oral gavage (~10^{12} bacteria/mouse) to mice ($n$ = 4) 24 h

following administration of 5 mg streptomycin (Fig. 3A). Colonisation remained high (Fig. 3B), and nearly all (>98%) barcodes and sensors (100%) were recovered on both days (Fig. 3C). Various sensors showed elevated switching in animals compared to gavage cultures (Fig. 3D; full dataset: Appendix Fig. S1). Consistent with

**Figure 3.  Barcoded biosensor screening of Library 1 successfully identifies biosensor responses to the healthy mammalian gut.**

(A) Library 1 was administered to C57Bl/6 mice 1-day post administration of 5 mg streptomycin ($n = 4$). Faecal samples were collected for colonisation and memory switching analyses the two days following. (B) Colonisation levels and (C) fractions of recovered barcodes and sensors showed high retention in all mice across both days. (D) Sensor activation of top-ranked in vivo responsive sensors, and control sensors in mice on days 1 ($n = 3$ passing QC) and 2 ($n = 4$ passing QC) post library administration. The heatmap shows global median from all data. The full dataset is shown in Appendix Fig. S1. (E) The response of the control sensor, Ec ynfE15 ($n = 3$ barcodes per gavage/animal), and top-ranked in vivo responsive sensor (F) Cr TCS7 (QseCB-P$_{ygiW}$) ($n = 6$-7 barcodes per gavage/animal) to the gut environment. Panels show FOR of individual barcode in faecal pellets from each mouse & gavage sample. Shades correspond to unique barcode variants with median response marked. Open circles indicate an outlier barcoded strain which was removed from further analyses. (G) DTR307 (Cr TCS7) was administered to streptomycin-treated mice ($n = 4$) and switching was monitored on the days following. (H) Percentage of activated DTR307 (Cr TCS7) colonies from plating of faecal contents. The graph shows counts from individual mice ($n = 4$) or gavage ($n = 1$) with mean marked. Statistics were calculated by one-sample $t$ test above the gavage switching value of 0.004%. Source data are available online for this figure.

previous testing (Naydich et al, 2019), Ec ynfE15-linked barcodes showed elevated FOR in the mouse gut, especially on day 2 (Fig. 3E).

The top-ranked gut-activated sensor was *Citrobacter rodentium* TCS7 (Fig. 3F), identified as the *ygiW* gene promoter paired with the adjacent and opposing TCS, QseCB. To validate the observed sensor response, we cloned Cr TCS7 as an individual strain (DTR307) and delivered it to mice via oral gavage 24 h following streptomycin treatment (5 mg) (Fig. 3G). DTR307 response to the gut was tested over the 48 h following administration via plating of faecal samples on indicator plates. Validating our library-based screening results, this sensor showed no evidence of switching in the gavage sample and elevated switching in bacteria collected on both days post gavage (mean switching: 16.9% day 1; 21.6% day 2) (Fig. 3H).

Another biosensor of interest identified by screening was Cr TCS2, the *C. rodentium citC* promoter with divergent TCS DpiAB (Fig. 3D). The 8 detectable Cr TCS2-linked barcodes showed a relatively low median response (<0.05 on both days), however, and high barcode-to-barcode variability in the mouse gut (Fig. EV3A). In vitro screening also suggested some response under general anaerobic growth conditions (Fig. 2A). The homologous TCS in *E. coli* is well-characterised, with induction by citrate, anaerobiosis and glucose and inhibition by nitrate. Bacteria may be exposed to all these conditions within the gut.

When cloned and tested in vitro as an individual sensor (DTR303), we confirmed strong response to citrate in the presence of glucose and anaerobic conditions (Fig. EV3B,C), strong inhibition by nitrate (Fig. EV3B) and occasional switching in anaerobic conditions even without further induction (Fig. EV3C). Truncation mutants further suggested that the *E. coli* host can regulate the heterologous promoter via DpiA (Fig. EV3D,E). To validate the screening results and test whether dietary citrate supplementation could enhance induction, DTR303 bacteria were administered via oral gavage to mice ($n = 4$–6 per group) fed streptomycin in their drinking water with or without the addition of 50 mM sodium citrate (Fig. EV3F). Colonisation levels were comparable between groups (Fig. EV3G). Indicator plating suggested switching of a small subset of sensor bacteria in some mice from each group, as sampled from both faecal (Fig. EV3H) and dissected gut contents (Fig. EV3I). Citrate supplementation did not clearly increase biosensor response above that seen in streptomycin-only treated mice (Fig. EV3H,I).

Taken together, these results demonstrate the ability for the screening pipeline to sensitively identify sensors of interest within the mouse gut and confirms that those with higher FOR values will likely correspond to higher switching percentages within the population.

## Identification and validation of inflammation-responsive biosensors

To specifically target inflammation-responsive circuit development, we combined Library 1 with Library 2—the latter including *E. coli* promoters previously associated with response in the monocolonised murine gut, including during inflammation (Schmidt et al, 2022). It remained unclear whether these promoters would facilitate inflammation-specific biosensing both in the presence of a complex, conventional microbiota and in the context of this memory circuit. We administered the combined libraries via oral gavage to mice ($n = 4$ per group; ~$10^{11}$ bacteria/mouse) pre-treated with streptomycin 24 h prior (5 mg/mouse) and with or without 3% dextran sulphate sodium (DSS) in their drinking water for the preceding 4 days to induce intestinal inflammation (Fig. 4A). Colonisation levels were high in all animals and similar across both groups (Fig. 4B). Reduction of colon length at endpoint and loss of body weight throughout the experiment confirmed the expected impacts of DSS in treated animals (Fig. 4C,D).

To account for the approximately tenfold increase in barcode abundance in the combined library compared with Library 1 alone (Fig. EV1C,D), increased volumes of faecal pellet supernatants (corresponding to 30–40 mg worth of faecal pellet per mouse, and approximately $10^7$ CFU of engineered bacteria as estimated from colonisation levels) were inoculated into outgrowths before splitting into ± spectinomycin cultures. Following barcode sequencing, fractional odds ratios were then calculated at the sensor level to account for the stochasticity of the lower abundance barcodes within the recovered fractions.

Several sensors showed elevated FOR in response to either the mouse gut or the DSS-treated gut specifically (Fig. 4E; full dataset: Appendix Fig. S2). The top 5 DSS responsive sensors included two *E. coli torC* promoter variants (with wt or synthetic 5'UTR MCD10), likely activated by trimethylamine-N-oxide (Baraquet et al, 2006); *E. coli yeaR* promoter, potentially activated by nitrate, nitrite or nitric oxide (Constantinidou et al, 2006); *E. coli spy* promoter, regulated by BaeR and CpxR (Kwon et al, 2010; Srivastava et al, 2014) and previously identified as being upregulated in response to DSS treatment in monocolonised mice (Schmidt et al, 2022); and, ST TCS1* (TtrRS-P$_{ttrBCA}$), previously implicated in tetrathionate sensing during inflammation (Riglar et al, 2017) (Fig. 4E–I). As expected, the control strain, Ec *ynfE15*, showed activation in all animals, with stronger sensor activation in

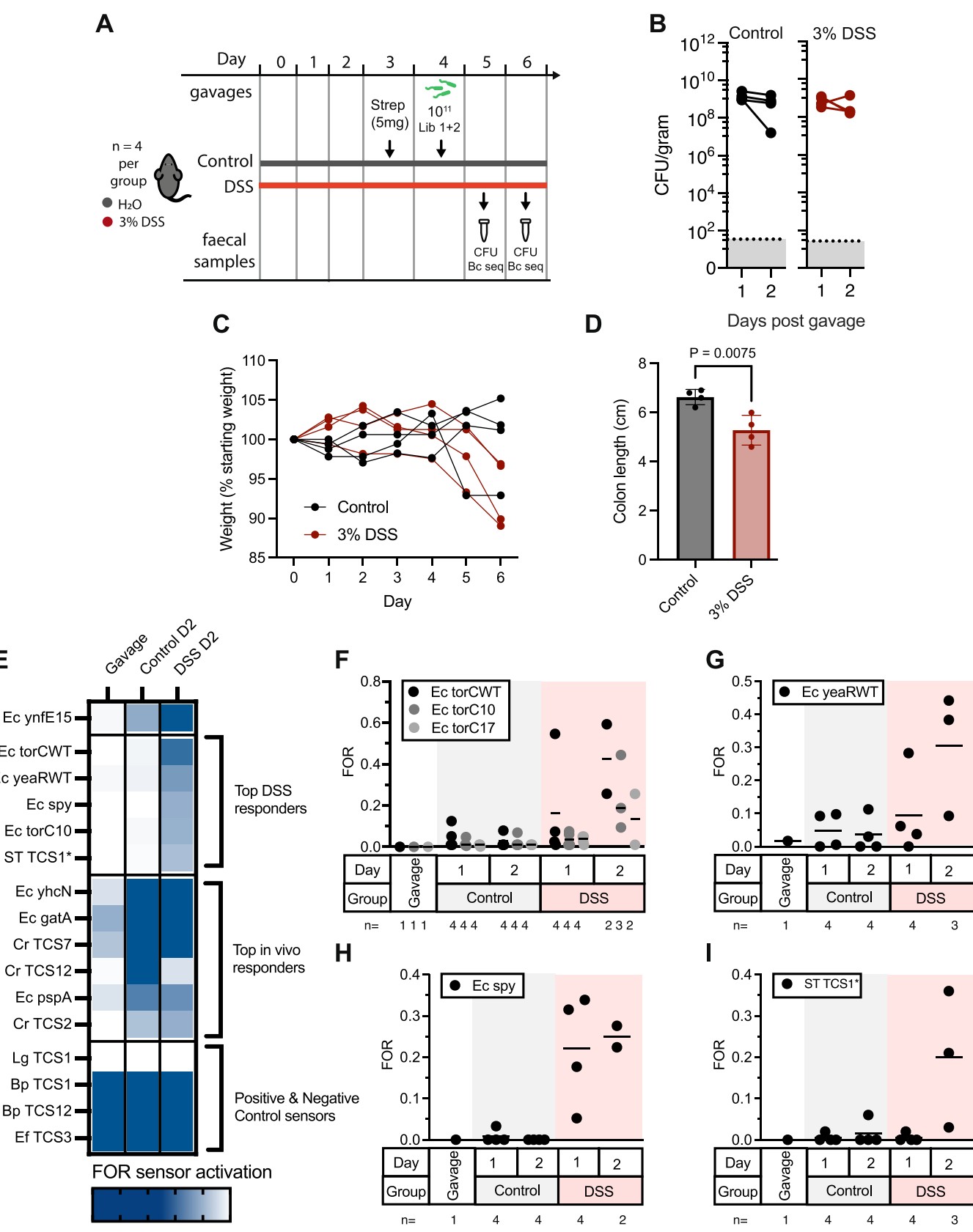

◄  **Figure 4.  Identification of inflammation-specific biosensors through screening of Library 1 + 2.**

(A) C57Bl/6 mice ($n = 4$ per group) were administered water with or without 3% DSS for 6 days to induce inflammation. At day 3, mice were dosed with 5 mg streptomycin and at day 4 with ~$10^{11}$ combined Library 1 + 2 sensors. (B) Faecal CFU counts on the 2 days following bacterial administration (day 5-6 post DSS exposure) ($n = 4$ per group) (C) Fractional weight change of mice and (D) colon lengths measured at the experimental endpoint on day 6 post DSS administration (day 2 post bacterial administration) ($n = 4$ per group). The graph shows mean ± standard deviation and results from an unpaired $T$ test. (E) Sensor activation of the top-ranked DSS responsive, in vivo responsive, and control sensors as averaged FOR across all samples passing QC on day 2 post library administration ($n = 1$ for Gavage; $n = 4$ for Control; $n = 3$ for DSS). The full dataset is shown in Appendix Fig. S2. (F–I) Individual sensor activation on day 1 and 2 post bacterial administration (day 5 and 6 post DSS exposure) for (F) Ec torC variants, (G) Ec yearWT, (H) Ec spy and (I) ST TCS1*. Graphs show individual mouse FOR values from datapoints passing sample and sensor QC criteria ($n = 1$ for gavage and $n = 2$–4 as labelled), with mean marked. Source data are available online for this figure.

faecal samples from day 2 (Figs. 4E and EV4A). Top-ranked sensors from Library 1 that activated in all animals again included Cr TCS7 (QseCB-P$_{ygiW}$) and Cr TCS2 (DpiAB-P$_{citC}$). Additional gut-responsive sensors of interest identified from Library 2 include the promoter region of *E. coli yhcN*, reported to be induced by cytoplasmic acid response (Kannan et al, 2008), Cr TCS12 (NarXL TCS-P$_{narK}$), and the promoter regions of *E. coli gatA* and *pspA* genes (Figs. 4E and EV4B–D).

Of the top-ranked inflammation-responsive sensors, Ec *spy* (Fig. 5A), showed particularly low response in control animals and consistent response across DSS-treated animals (Fig. 4H). To explore its induction behaviour in the context of our memory circuit more extensively, we therefore cloned and tested the sensor individually (DTR306). DTR306 cultures were grown for 4 h in aerobic SOC broth in the presence of previously reported inducers of the *spy* promoter: EtOH (5%), zinc (0.6 mM) and copper (0.6 mM) (Bury-Mone et al, 2009; Srivastava et al, 2014). Colony screening on indicator plates demonstrated clear induction of DTR306 by ethanol and zinc, but not copper (Fig. 5B). Neither did we see in vitro induction of DTR306 in the presence of 3% DSS, indicating the substance itself is not inducing sensor activation (Fig. 5B).

To confirm in vivo sensor behaviour, DTR306 was administered by oral gavage as an individual strain to mice. Mirroring the screening analysis, animals ($n = 4$ per group) were treated with or without 3% DSS in drinking water for six days (~$10^{11}$ bacteria/mouse). At day 3 post DSS treatment, streptomycin was provided (5 mg/mouse by oral gavage) 24 h prior to DTR306 administration (~$10^{11}$ bacteria per mouse) (Fig. 5C). Colonisation levels were high across the 2 days post administration in both groups (Fig. 5D). Weight loss throughout the experiment was indicative of the impact of DSS treatment (Fig. 5E). Average colon length was also reduced in DSS-treated mice, albeit without statistical significance (Fig. 5F). Analysis of DTR306 response via growth on indicator plates demonstrated a consistent increase in switching specifically in DSS-treated mice on both days tested (Fig. 5G). Growth of DTR306 from endpoint dissection of the gut also showed that switching was evident in all regions of the gut (Fig. 5H). Together, these results demonstrate activity of a synthetic biosensor capable of non-invasive monitoring and sustained synthetic circuit actuation throughout the gut with high specificity and sensitivity to the inflamed gut. Overall, they point to the high potential for this library-based construction and screening approach to identify diversely sourced inducers for the development of memory biosensors responsive to a multitude of conditions.

## Discussion

Our work expands the use of a robust bacterial memory circuit to simultaneously evaluate ~150 unique biosensors as a library of thousands of uniquely barcoded strains in the murine gut or in vitro culture. By incorporating a DNA barcoding strategy, using a single, hypervariable oligonucleotide for barcode delivery, we generate easily sequenceable libraries of sensors of diverse sizes ranging from a few hundred to several thousand base pairs in a low-cost, single-pot reaction. Cloning results in the simultaneous assembly of pooled sensors, each with multiple barcoded replicates. This facilitates assessment of sensor functionality and variability in each screen through measurement of replicate clones. The approach greatly reduces the cost and number of animals required for in vivo sensor testing and validation. For some identified sensors, barcodes revealed heterogenous response between replicates. This variability may derive from exposure to different microenvironments during transit of the non-uniform gut, unexpected mutations in a given sensor isolate, stochasticity in gene expression, and/or variations in strain outgrowth of the bacteria. No matter the cause, identification of response variability is an important factor in prioritising sensors for further development and assessment. Barcoding also facilitates iterative addition, combination, and revision of libraries over time, as demonstrated throughout this study. The successful identification and validation of several inducible circuits, both novel and previously identified, highlights the effectiveness of our pipeline in facilitating biotechnology tool development. The sensors tested across our study point to broad correlation between FOR levels measured during pooled screening and degree of response by an individually cloned sensor during validation (Fig. EV5).

The top hits for inflammation response (Figs. 4 and 5) are of interest for follow-up studies. Here, we have demonstrated the sensitivity and specificity of a memory biosensor driven by the *E. coli spy* promoter (Figs. 4 and 5). Spheroplast protein Y (Spy) is a small ATP-independent chaperone protein involved in the protein misfolding response (Kwon et al, 2010; Quan et al, 2011). Its induction, which is regulated by CpxAR and BaeSR TCSs (Bury-Mone et al, 2009; Yamamoto et al, 2008), includes response to envelope stressors such as ethanol (Fig. 5). *Spy* gene expression has been previously linked to DSS-induced inflammation in experiments with monocolonised mice (Schmidt et al, 2022), but this has not been validated in the presence of a conventional mouse microbiome. Several gut-relevant stimuli activate *E. coli* CpxAR, including high osmolarity (Jubelin et al, 2005), and exposure to antimicrobial peptides (Audrain et al, 2013). In *S.* Typhimurium

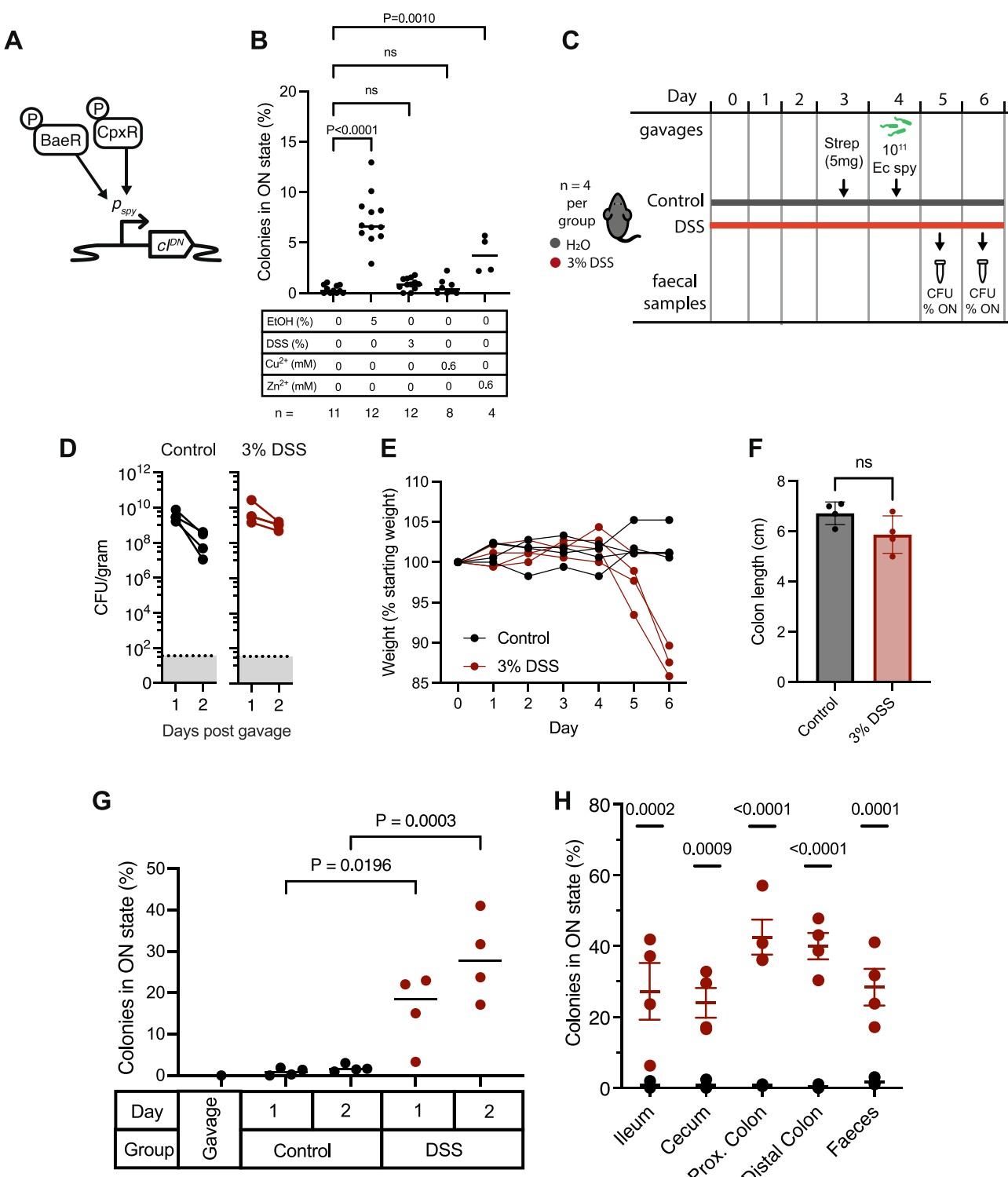

CpxAR is required for gut colonisation in an inflammation-dependent manner (Fujimoto et al, 2018). The high ranking of this and additional sensors with links to inflammation or inflammation-associated metabolites, including those based on designs from our previous studies with expected response to TMAO (through three *E. coli torC* sensor variants), nitrate/nitrite/nitric oxide (through *E.*

*coli yeaR*) and tetrathionate (through *S.* Typhimurium TtrRS-P_{ttrABC}) provides further confidence in our screening assay's potential (Riglar et al, 2017; Naydich et al, 2019). The strains provide a resource to longitudinally and non-invasively monitor these key aspects of the gut environment during inflammatory disease conditions via simple plating of faecal samples. New

**Figure 5.    Ec spy biosensor response in vitro and within the inflamed gut.**

(A) Predicted regulation of *E. coli* P$_{spy}$. (B) Response of the DTR306 (Ec spy) sensor when exposed to different inducers in vitro. Graphs show mean and individual counts from $n = 4$–12 biological replicates (as labelled). Statistics calculated by one-way ANOVA with Dunnet's multiple comparisons adjustment between groups as shown. (C) C57Bl/6 mice ($n = 4$ per group) with or without intestinal inflammation induced by 3% DSS administration were administered 5 mg streptomycin at day 3 and ~10$^{11}$ CFU DTR306 (Ec spy) bacteria at day 4 post DSS exposure. (D) Colonisation levels on the two days following sensor bacteria administration. (E) Fractional weight across the entire experiment and (F) colon lengths at endpoint on day 6 post DSS administration (day 2 post bacterial administration). P value: 0.0988 by unpaired *t* test. (G) Response of DTR306 (Ec spy) sensor by plating of faecal contents on days 1 and 2 post bacterial administration showed elevated response in DSS-treated animals. Graphs show counts from individual mice and mean. Statistics calculated by one-way ANOVA between groups as shown, with Šídák's multiple comparison adjustment. (H) Plating of contents from different intestinal regions at the experimental endpoint on day 2 post bacterial administration demonstrated elevated sensor memory response in all tested regions within the DSS-treated gut. Statistics calculated by two-way ANOVA with Šídák's multiple comparisons test between DSS and control groups at each location. Source data are available online for this figure.

biosensors identified and validated more broadly for activation within the gut environment (Figs. 3, 5, EV3 and 4) also provide important resources for the development of autonomously controlled synthetic circuits within the gut.

Novel sensor development is an area of need for synthetic biology in the quest to expand the toolbox of health- and disease-relevant inducible circuit components. Barcoded HTMS analysis is compatible with any transcriptional activation-based sensing circuits that function in the commensal *E. coli* host. This is a particular strength of our method, sourcing and testing diverse heterologous sensing components to drive memory circuit activity. The flexibility of Golden Gate assembly for allowing simultaneous assembly of multiple fragments, facilitates future expansion of the approach to more complicated sensor designs, for example, non-grouped TCSs, or those that regulate non-proximal or multiple promoters throughout the genome. Furthermore, the use of alternative type IIS restriction enzymes can be used to expand cloning compatibility of sensors, as demonstrated here during Library 2 construction which used both SapI and BsaI enzymes. Promising sensing components identified through HTMS-based screening can be optimised for this or other transcriptional outputs, including through dynamic tuning, which can be achieved, for example, through RBS modulation or the addition of genetic amplifier components (Wan et al, 2019).

The cloning strategy afforded libraries of hundreds (Library 1) or thousands (Library 2) of uniquely barcoded strains, tagging ~150 variant sensors, including TCS and promoter-based sensors. Not all targeted sensors were successfully cloned, with lack of PCR amplification the greatest source of loss within the pipeline. This is indicative of the challenges of undertaking PCR primer design and reactions in bulk using diverse bacterial genomes. Of the PCR-amplified sensors, >80% were successfully assigned to at least 1 barcode in the library. The remaining 20% may be absent based on stochastic abundance variations in the amplification and cloning process but could also result from toxicity or burden caused by the heterologous genes when introduced into *E. coli*, or differences in Golden Gate assembly efficiency during the pooled multiplexed library construction.

A fraction of the sensors were constitutively ON in all tests, which is expected in a memory system for any gene promoters that are routinely activated during in vitro culture conditions. In some cases, this may also result from a lack of native regulation and thus high basal expression of cI$^{DN}$ in the heterologous host. TCSs derived from species more closely related to the *E. coli* NGF-1 host were,

unsurprisingly, more likely to show signs of functionality, consistent with systematic assessment of porting non-native genetic systems into new hosts (Wang et al, 2019). Many promoters are controlled by combinations of TCSs, and most TCSs regulate gene expression from a variety of targets (Rajeev et al, 2020), suggesting that cross-regulation between host and heterologous TCSs and gene expression is both likely and necessary. For example, tetrathionate sensing by ST TCS1* and Et TCS7 requires cross-regulation by host *E. coli* FNR (Fig. 2). Our experiments with truncated *Cr* TCS2 sensor variants also demonstrated cross-regulation by host *E. coli* DpiAB, NarL and cAMP:CRP (Fig. EV3). Although many promoters were selected rationally based on prior evidence of upregulation or essentiality during inflammation, screening identified only a handful with promise as strong and specific sensors when used in this context within the conventional mouse gut. Together, these challenges point to the power of library screening approaches to accelerate the identification of promising sensors from larger pools to account for high sensor failure rates.

During in vivo library screening experiments, population bottlenecking caused barcode and strain diversity loss following administration to the gut, ultimately limiting experiment length (Fig. EV2). Such colonisation bottlenecks are consistent with work examining barcoded pathogenic strains in the mouse gut (Campbell et al, 2023; Woodward et al, 2023), suggesting this may be a fundamental limitation to library screening approaches on the background of a complete microbiota. Nevertheless, the success of our approach suggests that library screening approaches are still feasible and valuable over short experimental periods (e.g., up to 2 days). Testing directly in the λ-memory system encoded in mouse commensal *E. coli* NGF-1, which has previously demonstrated low cellular burden, long-term stability, and function over many months in the murine gut (Kotula et al, 2014; Naydich et al, 2019; Riglar et al, 2017) also allows for rapid progression from short-term screening to longer-term follow-up tests as a single sensor, where these bottlenecking issues are not a concern.

This pipeline not only enables the construction of biosensors tailored to the gut environment but also offers the potential for screening biosensors in other microbial growth-permissible and potentially inaccessible environments, such as within microbial biofilms, tumours, or enclosed artificial systems such as bioreactors or microfluidic systems. The ability to build and test new whole-cell biosensors at high throughput is a powerful method for the identification of novel inducible synthetic circuit components and prototype disease diagnostics.

# Methods

### Reagents and tools table

| Reagent/resource | Reference or source | Identifier or catalogue number |
| --- | --- | --- |
| **Experimental models** | | |
| C57BL/6J female mice (7–9 weeks old) | Charles River UK | N/A |
| **Strains** | | |
| ElectroMAX DH5a-E competent cells | Thermo Fisher Scientific | 11319019 |
| Bacterial strains (various) | Various | Table EV2 |
| **Recombinant DNA** | | |
| pDR07 | Naydich, 2019, mSystems | N/A |
| pCR06 | This study | N/A |
| Nextera™ Compatible Indexing Primers (UDI 10 bp) | Integrated DNA Technologies | N/A |
| **Oligonucleotides and other sequence-based reagents** | | |
| *Edwardsiella tarda* purified gDNA | Public Health England | NCTC 10396 |
| *Blautia producta* purified gDNA | Public Health England | NCTC 11829 |
| PCR primers and DNA fragments | This study | Table EV1 |
| Nextera™ Compatible Indexing Primers (UDI 10 bp) | Integrated DNA Technologies | N/A |
| **Chemicals, enzymes and other reagents** | | |
| Tryptone | Sigma-Aldrich | T7293-250G |
| Yeast Extract | Thermo scientific | MgCl2 |
| NaCl | Sigma-Aldrich | S988 |
| KCl | Sigma-Aldrich | P9541 |
| MgCl$_2$ | Merck | M8266 |
| MgSO$_4$ | Sigma-Aldrich | 83266 |
| Glucose | Fisher Scientific | FSUG0500 |
| Triton X-100 | Roche | 10789704001 |
| Tris HCl | Fisher Scientific | J22638 |
| Sodium EDTA | Invitrogen | AM9260G |
| Lysozyme | Sigma-Aldrich, | 62971 |
| Sucrose | VWR | 0335 |
| Agar | Thermo Scientific | 443572500 |
| PBS | Fisher Chemical | 1282-1680 |
| LB broth (Miller) | VWR | 84649 |
| Peptone special | Sigma-Aldrich | 68971 |
| Potassium tetrathionate | Sigma-Aldrich | P2926 |
| Sodium citrate tribasic dihydrate (used for bacterial induction experiments) | Sigma-Aldrich | 71406 |
| Sodium citrate (food grade, used for mouse administration) | Classikool Ltd | E331 |
| Sodium nitrate | VWR | 27955 |
| Copper chloride dihydrate | Sigma-Aldrich | C6641 |

| Reagent/resource | Reference or source | Identifier or catalogue number |
| --- | --- | --- |
| Zinc sulphate | Sigma-Aldrich | Z0251 |
| M9 Difco M9 Minimal Salts | Scientific Laboratory Supplies Ltd | 248510 |
| Casamino acids | Fisher Scientific | 12747109 |
| Niacinamide | Merck | N0636 |
| Q5 High-Fidelity DNA polymerase | New England Biolabs | M0491S |
| Gentra Puregene Yeast/ Bacteria gDNA extraction kit | Qiagen | 158567 |
| Monarch PCR Cleanup Kit | New England Biolabs | T1130 |
| Monarch Plasmid Miniprep Kit | New England Biolabs | T1010 |
| SapI | New England Biolabs | R3733 |
| BsaI | New England Biolabs | R3733 |
| T4 DNA Ligase | New England Biolabs | M0202 |
| T4 DNA Ligase buffer | New England Biolabs | B0202 |
| Nanopore Ligation Sequencing Kit | Oxford Nanopore Technologies | SQK-LSK109 |
| Flongle flow cell (R9.4.1) | Oxford Nanopore Technologies | FLO-FLG001 |
| Minion flow cell (R10.4.1) | Oxford Nanopore Technologies | FLO-MIN114 |
| innuPREP DNA/RNA kit | Analytik Jena | 845-KS-2080250 |
| AMPure XP beads | Beckman Coulter | A63880 |
| Qubit dsDNA BR assay kit | Invitrogen | Q32850 |
| Teklad 2019 Global Rodent Diet | Envigo | 2019 |
| Streptomycin sulphate (used for mouse administration) | VWR | 0382-EU-50G |
| Streptomycin (used for bacterial culturing) | Sigma-Aldrich | S650 |
| Dextran sodium sulphate | MP Biomedicals | 160110 |
| X-gal (5-bromo-4-chloro-3-indolyl-b-d-galactopyranoside) | Thermo Fisher Scientific | R0402 |
| Chloramphenicol | Sigma-Aldrich | C0378 |
| Carbenicillin | ThermoFisher Scientific | BP2648 |
| Spectinomycin | MP Biomedical | 158993 |
| **Software** | | |
| TCS extraction pipeline | https://github.com/riglarlab/Robinson_BiosensorsLibrary | |
| Geneious Prime version 2020.1.1-2023.1.1 | Biomatters Ltd | |
| MinKNOW (version 22.12.5) | Oxford Nanopore Technologies | |
| Minimap2 (version 2.17) | https://github.com/lh3/minimap2 | |
| Bowtie2 | https://github.com/BenLangmead/bowtie2 | |
| Dada2 | https://benjjneb.github.io/dada2/ | |

| Reagent/resource | Reference or source | Identifier or catalogue number |
|---|---|---|
| Graphpad Prism v9 & v10 | https://www.graphpad.com/ | |
| **Other** | | |
| CyBio FeliX liquid handling robot | Analytik Jena | |
| BD GasPak EZ container system with container system sachets | Becton Dickinson | 260001 |
| InvivO$_2$ workstation with I-CO$_2$N$_2$IC gas controller fitted for anoxic conditions | Baker Ruskinn | |
| GridION Nanopore Sequencer | Oxford Nanopore Technologies | |

## Methods and protocols

### TCS Library construction

Additional oligonucleotide and primer sequences used for library construction and sequencing (Table EV1) and bacterial strains used throughout the study (Table EV2) are provided.

Purified bacterial gDNA was sourced from Public Health England (NCTC 10396 *Edwardsiella tarda* and NCTC 11829 *Blautia producta*), by donation from other laboratories, or, for *Citrobacter rodentium*, purified using the Gentra Puregene Yeast/ Bacteria gDNA extraction kit (Qiagen). TCSs were amplified by PCR using Q5 High-Fidelity DNA polymerase (New England Biolabs), and amplified PCR fragments were purified by column purification using the Monarch PCR Cleanup Kit (New England Biolabs) following the manufacturer's protocols.

DNA barcodes were synthesised as an Ultramer DNA Oligo (IDT) consisting of a 106 bp variable nucleotide region flanked by Illumina sequencing primer binding sites, allowing direct sequencing of the barcode region, and SapI restriction sites, for Golden Gate Assembly (GGA) (Table EV1). The DNA barcode oligo was made double-stranded by Q5 PCR with 5 cycles with 8 pmol of starting oligo template and 32 pmol of each duplexing primer (primers p070 and p075 for SapI-based GGA, and p079 and p100 for BsaI-based GGA, in Table EV1).

The destination vector was pDR07 (for BsaI assembly) (Naydich et al, 2019) or pCR06 (for SapI assembly), both modified Tn7 transposon backbone derived from pGRG36 (McKenzie and Craig, 2006). Amplified sensors were pooled and assembled with the double-stranded barcode oligo into the Tn7 transposon machinery containing backbone (pCR06 or pDR07) by multiplexed GGA using SapI or BsaI. SapI GGA reactions were set up using: 3 nM of each DNA fragment, 3 μL(15 units) of SapI, 0.25 μL(500 units) of T4 DNA Ligase, 2 μL of T4 DNA Ligase buffer and H$_2$O to a final volume of 20 μL. BsaI GGA reactions were set up using: 3 nM of each DNA fragment, 1.5 μL (30 units) of BsaI-HFv2, 0.5 μL (1000 units) of T4 DNA Ligase, 2.5 μL of T4 DNA Ligase buffer and H$_2$O to a final volume of 25 μL. Cycling conditions for all GGA were: (37 °C, 5 min and 16 °C, 5 min) × 30, followed by 60 °C, 5 min, then storage at 4 °C. For each library, 4 μL of the GGA mix was transformed into 50 μL of ElectroMAX DH5α-E competent cells by electroporation according to the supplier's protocol and plated onto

5 carbenicillin-selective LB-agar plates. An additional 1:100 dilution of the transformation reaction was plated to estimate library size. All transformants were scraped from LB-agar transformant plates using 1 mL of pre-warmed LB, vortexed to mix thoroughly, and frozen as a glycerol stock. A large inoculum of the glycerol stock was grown in 5 mL of LB overnight, miniprepped and transformed into electrocompetent *E. coli* NGF-1 encoding HTMS memory (PAS811). Transformant plates were scraped with pre-warmed LB, pooled, diluted 1:1000 and grown for 6 h at 30 °C in LBPS–streptomycin–carbenicillin–chloramphenicol with 0.1% L-arabinose to induce Tn7-based genome integration. Cells were then diluted 1:1000 and grown overnight at 42 °C three times for plasmid curing. Libraries were frozen as 25% glycerol stocks until use.

Individually cloned sensors (Ec yeaRWT, Ec yeaR23, Ec ynfE17, Ec torC10, Ec torC17, Ec torC23, Ec torCWT, Ec hycAWT), control sensors (Ec ynfE15 and ST TCS1*), and fabR positive normalisation strains were cloned using identical protocols to the library construction, with the exception of using individual sensor PCRs for GGA reactions (Dataset EV1) and picking multiple separate (so separately barcoded) individual colonies for plasmid integration and curing steps. Sensors and barcodes of individually cloned constructs were verified and identified by Sanger sequencing (Eurofins) (Dataset EV1). Correctly assembled, individually cloned sensors were combined with the pooled library at ~1:1000.

### Truncation mutant cloning

DpiA-binding sites within the Cr TCS2 intergenic promoter region were identified by sequence alignment to the corresponding genomic region in *E. coli* MG1655 within the Ecocyc database (Keseler et al, 2021). Plasmids for truncated sensor mutants DTR304 and DTR305 were cloned by PCR amplifying appropriate regions (using DTR304_Fwd and DTR304_5_Rev or DTR305_Fwd and DTR304_5_Rev, respectively), assembled by GGA using SapI along with unique barcodes, and transformed and genome-integrated into PAS811 as above.

### Nanopore sequencing and barcode assignment

For nanopore long-read sequencing, the barcode and sensor regions of the biosensor libraries were amplified using Q5 PCR (New England Biolabs) directly off 5 μL of glycerol stock using primers p117 and p118 (Table EV1). PCR products were purified using a Monarch PCR Cleanup Kit (New England Biolabs) and prepared for nanopore long-read sequencing using the Ligation Sequencing Kit (Oxford Nanopore Technologies, SQK-LSK109). Library 1 was run on a Flongle flow cell (R9.4.1) and Library 2 was run on a Minion flow cell (R10.4.1). Base calling from Fast5 files was completed using MinKNOW software (version 22.12.5). Nanopore adapters were trimmed in Geneious (adapter sequence: CCTGTACTTCGTTCAGTTACGTATTGCT) with 5 bp allowed mismatches, minimum 10 bp match to the adapter and 0.1 error probability limit (regions with more than a 10% chance of error per base were trimmed).

Trimmed nanopore data was filtered for reads >400 bp (a read length sufficient to at minimum partially cover both the barcode and sensor regions), aligned to TCS reference sequences (excluding barcode regions) of all successfully PCR-amplified TCS using Minimap2 (version 2.17). Long-read base pairs that extend past the TCS reference sequence were trimmed using a custom Python

script and aligned to the inferred barcodes from short-read sequencing data (see below) using Bowtie2 (Langmead and Salzberg, 2012). Barcodes were assigned to a specific sensor (Dataset EV1) if that sensor had both the highest number of trimmed long reads aligning to the barcode, and highest fraction of aligning trimmed long reads normalised by total long reads for the specific sensor, with a minimum of 5 long reads for assignment. Any barcodes with different assignments between those values, or to which no trimmed long-read data aligned, were discarded for odds ratio calculations.

### gDNA extraction

For short-read sequencing, genomic DNA (gDNA) was extracted from ± spectinomycin overnight cultures manually using the Gentra Puregene Yeast/Bacteria gDNA extraction kit (Qiagen catalogue no. 158567) or on a FeliX liquid handler using the innuPREP DNA/RNA kit with an initial manual lysozyme step for bacterial cell lysis. For the lysozyme bacterial cell lysis steps, 500 µL of bacterial overnight culture was pelleted, resuspended in 200 µL of enzymatic buffer consisting of 20 mM TrisCl pH 8, 2 mM sodium EDTA, 1.2% Triton X-100, and 20 mg/mL lysozyme and incubated at 37 °C for 30 min.

### Biosensor testing

Antibiotics in media for bacterial growth were used at the following working concentrations: streptomycin 200 µg/mL, chloramphenicol 25 µg/mL, and spectinomycin 50 µg/mL. Unless otherwise stated, all bacterial growth steps were done at 37 °C in either a shaking incubator for liquid cultures (200 RPM for 14-mL round bottom culture tubes, 1200 RPM for 96-deep well plates), static incubator for growth on LB-agar plates, or statically under anaerobic conditions using either a BD GasPak EZ container system or using an InvivO$_2$ workstation with I-CO$_2$N$_2$IC gas controller fitted for anoxic conditions.

### Individual biosensor testing in vitro

1. Prepare SOC media containing: 2% Tryptone, 0.5% yeast extract, 10 mM NaCl, 2.5 mM KCl, 10 mM MgCl$_2$, 10 mM MgSO$_4$, 20 mM glucose. For induction, add the appropriate inducer (e.g. potassium tetrathionate, sodium citrate, sodium nitrate, etc.) to the SOC media at the required concentration for induction. Prepare X-gal LB-agar plates by adding 60 µg/mL X-gal to warmed, sterile LB-agar before pouring into petri/bacterial culturing plates and allowing to solidify.
2. Inoculate a single colony of the biosensor strain in streptomycin-chloramphenicol-LB media and grow overnight aerobically.
3. Dilute the biosensor strain by a factor of 1:1000 into the prepared SOC media with inducer. If testing anaerobic induction, SOC media with the inducer must be incubated under anaerobic conditions for at least 24 h before biosensor inoculation to ensure adequate removal of oxygen from the media. If testing multiple conditions simultaneously, this can be done in a 96-deep-well plate.
4. Grow the biosensor strain in the SOC media with inducer for 4 h, then dilute and spread on X-gal streptomycin-chloramphenicol-LB-agar plates at a bacterial concentration that allows for resolution of individual colonies (usually a dilution of approximately 1:10$^5$) and incubate overnight.

5. Count blue and white bacterial colonies to quantify biosensor activation.

### High-throughput in vitro library testing with tetrathionate

1. Prepare LB peptone special (LBPS) containing: 10 g/L peptone special, 10 g/L NaCl, and 5 g/L yeast extract in deionized water.
2. Inoculate the following strains or biosensor library into 4 mL of streptomycin-LBPS and grow overnight
   a. Approximately 15 µL of a bacterial biosensor library glycerol stock
   b. fabR barcoded strains (DTR270-273)
3. Add the appropriate inducer (potassium tetrathionate) to streptomycin-LBPS at the required concentration for induction.
4. Dilute the biosensor library by a factor of 1:1000 into the prepared LBPS media with inducer, and grow for 4 h in aerobic or anaerobic conditions.
5. Dilute the barcoded fabR strains (DTR270-273) by 1:100 into the induced biosensor library cultures, then dilute the library+fabR strain mixed cultures by 1:100 into fresh streptomycin-LBPS media ± spectinomycin and grow overnight aerobically.
6. Centrifuge the cultures to obtain cell pellets and extract gDNA using either the Gentra Puregene Yeast/Bacteria gDNA extraction kit or the innuPREP DNA/RNA kit and a CyBio FeliX liquid handling robot.
7. Biosensor activation can then be quantified following the 'short read sequencing barcode analysis and fractional odds ratio calculation, and sample QC' protocol section.

### High-throughput in vivo library screening and screening of individual strains in vivo

All animal experiments were carried out under approval of the local Ethical Review Committee at Imperial College London according to UK Home Office guidelines (PPL PP7088487). Experiments were conducted using 7–9 week old C57BL/6J female mice (Charles River UK). Upon arrival at the facility mice were randomised to cages according to the required group size, taking individual animals as the experimental unit, then given one week to acclimatise to housing conditions and diet. While formal power calculations were not possible for the experiments undertaken, group sizes were informed by both the equations of Dell and colleagues (2002) and prior experience. Cages were randomly assigned to control and treatment groups where relevant. No other confounders were formally controlled. No animals were entirely excluded from study, however, during blinded analysis individual samples were excluded in some cases if they didn't meet the required criteria described in the sample QC section and individual sensors were not analysed from the library if insufficient reads were recovered. All animals were sacrificed humanely by cervical dislocation at the endpoint of the experiments. Mice were provided with Teklad 2019 Global Rodent Diet (Envigo). For all faecal pellet collections mice were temporarily moved into clean, sterile boxes from which pellets were directly collected and placed into 100 µL PBS.

1. Pre-treatment (24 h before biosensor library administration)
   a. Administer streptomycin by oral gavage (0, 1, or 5 mg as specified) in 100 µL PBS. For DTR303 (Cr TCS2) testing, one day before administration of bacteria, the citrate induced

group were provided 50 mM sodium citrate with 5% sucrose in their drinking water and this was continued throughout the experiment.

b. Confirm the absence of streptomycin-resistant bacteria 24 h post streptomycin administration by collecting fresh faecal pellets and plating on streptomycin selective LB-agar plates.

2. Biosensor (single strain or library) administration

a. Inoculate ~15 μL of a glycerol stock (for biosensor libraries) or a single colony (individual biosensors, e.g. DTR306, DTR303) into 4 mL of streptomycin-LBPS and grow aerobically overnight.

b. Centrifuge the cultures to pellet the cells, remove media supernatant, and resuspend the cell pellet in 4 mL of PBS to wash the cells once, then repeat pelleting and removal of supernatant and resuspend in 400 μL (Library 1 testing) or 800 μL (Library 1 + 2 testing and individual strain testing) of PBS to concentrate the bacterial cultures by 10× or 5×.

c. Administer 200 μL of the washed and concentrated libraries or individual sensors by oral gavage (~$10^{10-12}$ cells per mouse).

3. Sensor activation and bacterial load quantification

a. At desired timepoints collect fresh faecal pellets and store them immediately in Eppendorf tubes with 100 μL of PBS, then add an appropriate volume of PBS to reach a concentration of 100 mg/mL (measure the weight of the Eppendorf tubes before and after addition of the faecal pellet to calculate the weight of the faecal pellet and the correct volume of PBS required).

b. Vortex samples for 2 min, then pulse centrifuge (<2 s) to remove large debris.

c. Create several dilutions of the faecal slurries and spread on X-gal-streptomycin-chloramphenicol-LB-agar indicator plates. Grow overnight and count the number of biosensor colonies to track faecal bacterial load or, for individual sensor testing, blue and white colonies to quantify sensor activation.

d. For DTR306 (Ec spy), to test sensor activation within different regions of the gut, at the endpoint of the experiment mice were dissected and the entire contents including mucus of the small intestine, caecum, proximal colon or distal colon were scraped, collected, resuspended to 100 mg/mL in PBS, vortexed, centrifuged, diluted and plated as for faecal pellets. The proximal colon was selected as the top half of the colon (from the bottom of the caecum to halfway to the anus), and distal colon is defined as the bottom half of this region.

e. For library-based biosensor activation, inoculate the remaining faecal supernatant (after dilution) into 4 mL of streptomycin-chloramphenicol-LBPS and grow for 4 h.

f. After 4 h, further dilute into 1:100 into 10 mL fresh LBPS, and split into 2× 5 mL culture. Add spectinomycin to one culture and grow both overnight.

i. Note: For testing Library 1, positive normalisation fabR strains were also diluted 1:1000 from overnight cultures and grown for 4 h, alongside faecal pellet outgrowth samples, before being diluted by 1:100 into the faecal pellet outgrowths and split into ± spectinomycin cultures for overnight growth.

g. Centrifuge the cultures to obtain cell pellets and extract gDNA using either the Gentra Puregene Yeast/Bacteria gDNA extraction kit or the innuPREP DNA/RNA kit and a CyBio FeliX liquid handling robot.

h. Biosensor activation can then be quantified following the 'short read sequencing barcode analysis and fractional odds ratio calculation and sample QC' protocol section.

### Short-read sequencing barcode analysis and fractional odds ratio calculation, and sample QC

1. Barcode amplification for short-read sequencing:

a. Amplify barcodes from genomic DNA using Q5 PCR with primers p145 and p146 (Table EV1) and purify the PCR products using AMPure XP beads at 1.8x beads to sample ratio following the manufacturer's protocol (Beckman Coulter).

b. Perform second Q5 PCR amplification using IDT Nextera™ Compatible Indexing Primers (UDI 10 bp) with a modified protocol of 12 amplification cycles, then purify again using AMPure XP beads as in step a.

c. Quantify final fragments using Qubit dsDNA BR assay kit following the manufacturer's protocol and pool samples at equimolar concentrations.

d. Sequence on Illumina NovaSeq using 150 bp Paired-end mode, read depth for this work was at an average of 1.4 million reads per sample.

2. Barcode short-read sequencing data analysis:

a. Infer barcode counts using Dada2 with the following settings: truncLen=106, maxEE=1, truncQ=2. Inferred barcodes were further filtered to the expected barcode length of 106 bp and clustered on 95% sequence similarity using the seq_cluster function from bioseq. For this work, the exact barcode DNA sequence with the highest read number in each cluster was kept as the cluster's reference barcode and read counts from other barcode members of the cluster added to the reference barcode.

3. Fractional odds ratio (FOR) calculation

a. Calculate sensor activation using a fractional odds ratio defined for each sensor. First, calculate the odds ratio for each barcode in a sample as

i. $OR_{sensor} = \dfrac{BC_{+SPECT}/mean(PNB_{+SPECT})}{BC_{-SPECT}/mean(PNB_{-SPECT})}$

ii. Where $BC_{+SPECT}$ and $BC_{-SPECT}$ are reads from an individual barcodes and $PNB_{+SPECT}$ and $PNB_{-SPECT}$ are combined read numbers from several positive normalisation barcodes in samples grown with and without spectinomycin, respectively.

iii. Note: Odds ratios were initially calculated using the averaged results from four barcoded strains triggered by our previously used 100% positive normalisation promoter from *E. coli* MG1655, pFabR (Naydich et al, 2019). However, in certain samples, we observed variability in the outgrowth of spiked-in strains compared to sensors contained within the library, which appeared to grow more slowly (Appendix Fig. S3A-B). We hypothesised this may result from the differential growth conditions of normalisation strains and library prior to mixing, leading to differential lag phases prior to growth. Thus, we developed a more robust normalisation strategy using 14 barcodes linked to 3 sensors contained within the library (Bp TCS1, Bp TCS12 and Ef TCS3) that consistently showed odds ratios ~1 when normalised using fabR strains, and which were subsequently validated as constitutively switching ON

by independent cloning and indicator plating (Appendix Figure S3C). When samples were normalised using these barcodes instead of the fabR barcoded strains, it corrected skewness that could lead to shifted odds ratio (Appendix Fig. S3D).

b. Using the OR values, calculate the fractional OR (FOR):

 i. $FOR = \frac{OR_{sensor} - OR_{neg}}{OR_{pos} - OR_{neg}}$

 ii. Where ORpos is the average odds ratio of all positive control strains (Ef TCS3, Bp TCS12 and Bp TCS1), and ORneg is the average odds ratio of an internally selected negative control (OFF) strain. Lg TCS1 was selected as an internal negative control, as it was the sensor with >5 assigned barcodes that consistently showed the lowest odds ratios across all analysed samples.

 iii. Note: FOR values as opposed to OR values are used to calculate sensor activation as because spectinomycin is bacteriostatic, reads from OFF sensors may remain in the culture at an abundance determined by the fold change of the population's outgrowth. This will vary based on the dilution factor used (typically 1:1000 for in vitro data, 1:100 for in vivo data) and *E. coli* cell density in the original sample (which is variable).

4. Sample QC:

Note: Due to the roughly 10-fold higher number of barcodes in Library 2, reads from all barcodes for a specific sensor in these samples were pooled before OR, FOR and QC calculations.

a. Outgrowth variability QC:

 i. Calculate Slope and $R^2$ of a linear fit of OR of positive control strains to exclude samples with high levels of observed outgrowth variation (thresholds set at an $R^2 > 0.6$ and slope > 0.5 and < 2).

 ii. Note: Replicate barcodes of individual sensors within the library allowed assessment of odds ratio variability caused by differences in bacterial starting concentration. Low concentration samples display higher variability in their outgrowth, resulting in higher deviation from the expected odds ratio of 1 for the 14 positive barcodes (Appendix Fig. S4).

b. Barcode abundance QC:

 i. Filter all barcodes to be above a set threshold of an abundance in the final sequenced-spectinomycin sample of >1/5000 of the fraction of reads. To avoid any impacts from low-abundance positive normalisation barcodes, positive control barcodes with <100 reads in ± spectinomycin cultures from a given sample were excluded.

 ii. Note: The libraries contain sensors with abundances across ranges of orders of magnitude. The threshold of abundance >1/5000 is because fractions below this threshold would indicate tens to a couple of hundred bacterial cells being split between +spectinomycin and −spectinomycin cultures (from a stationary phase *E. coli* culture containing $10^8 - 10^9$ CFU/mL, a sensor at 1/5000 of the library is at a concentration of around $2*10^4 - 2*10^5$ CFU/mL, after a 1:1000 dilution into ± spectinomycin cultures would be at 20 − 200 CFU/mL).

c. Fold change QC:

 i. Calculate the fold change in the OR of negative to positive control strains (by using the geometric mean of the barcodes of negative or positive strains mentioned in 3.b.ii.) (threshold set at FC > 30).

 ii. Note: Insufficient difference between positive and negative strains indicates there was insufficient outgrowth of the library to reliably quantify ON or OFF sensors.

### Statistics and sensor rankings

Statistical tests were performed in GraphPad Prism v9 or v10 software. Details of specific tests performed are included in the relevant figure legends. For ranking responsive sensors in the Library 1 in vivo screen (Fig. 3D) and Library 1 + 2 in vivo screen (Fig. 4E), sensors were first filtered for those OFF in the gavage sample (FOR < 0.25). To rank in vivo responsive sensors for Library 1 screening (Fig. 3), sensors were ranked by the greatest difference between average faecal FOR across all mouse samples and gavage FOR. To rank inflammation-responsive sensors in the Library 1 + 2 screen (Fig. 4), sensors were ranked by the difference in average faecal FOR in DSS-treated animals (D2) and average faecal FOR in control animals (D2).

## Data availability

The datasets and computer code produced in this study are available in the following databases or files: Raw sequencing data: European Nucleotide Archive project number PRJEB66358; Genetic construct DNA sequences: TCS DNA sequences, including intergenic regions up to neighbouring genes and primer sequences used to amplify the TCS, and library barcode sequences are available in Dataset EV1; Sequence data analysis scripts: Github (https://github.com/riglarlab/Robinson_BiosensorsLibrary); Strains are available by contacting the corresponding author upon reasonable request.

The source data of this paper are collected in the following database record: biostudies:S-SCDT-10_1038-S44320-025-00123-3.

## Peer review information

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

## Acknowledgements

The authors would like to thank R Jackson, M Crone, N Matthews, and K Rekopoulou for assistance with sequencing and sequencing analysis; M Tran and K Jensen for assistance with automation; G Frankel, J Marchesi, C Mullineaux Sanders, L Roberts and D Chrystomou, for gifts of, and assistance with, genomic DNA. PAS811 was a gift from P Silver. Funding was provided by the Wellcome Trust and Royal Society Sir Henry Dale Fellowship 211230/Z/18/Z (DTR) and Imperial College London President's PhD Scholarship (CMR).

## Author contributions

**Clare M Robinson**: Conceptualisation; Data curation; Software; Formal analysis; Funding acquisition; Investigation; Visualisation; Methodology; Writing—original draft; Writing—review and editing. **David Carreño**: Conceptualisation; Investigation; Methodology; Writing—review and editing. **Tim Weber**: Investigation; Methodology; Writing—review and editing. **Yangyumeng Chen**: Investigation; Writing—review and editing. **David T Riglar**: Conceptualisation; Resources; Formal analysis; Supervision; Funding acquisition; Investigation; Visualisation; Methodology; Writing—original draft; Project administration; Writing—review and editing.

Source data underlying figure panels in this paper may have individual authorship assigned. Where available, figure panel/source data authorship is listed in the following database record: biostudies:S-SCDT-10_1038-S44320-025-00123-3.

## Disclosure and competing interests statement

The authors declare no competing interests.

# Expanded View Figures

**Figure EV1. Library cloning strategy and validation.**

(**A**) Libraries were constructed by multiplexed golden gate assembly to create a barcoded plasmid library, which was first transformed into *E. coli* DH5a, then miniprepped and transformed into *E. coli* NGF-1 (PAS811), a murine gut commensal *E. coli* strain carrying the high-throughput memory circuit. Induction of Tn7 genome integration machinery by arabinose and curing of plasmids by multiple growth steps at 42 °C afforded each final pooled library. (**B**) Barcodes were assigned to sensors by a combination of Nanopore long-read sequencing and Illumina short-read sequencing. Lower fidelity long-read data were first aligned to sensor templates, and associated barcode regions were then trimmed and aligned to high confidence barcode sequences that were inferred using Dada2 clustering from Illumina short-read sequences. A barcode was assigned to a sensor if it had both the highest number of trimmed long reads and highest fraction of aligning trimmed long reads normalised by total long reads for the specific sensor. (**C**) Number of assigned barcodes per sensor within libraries 1 and 2. (**D**) Fractions of total reads across all samples of each library's assigned barcodes.

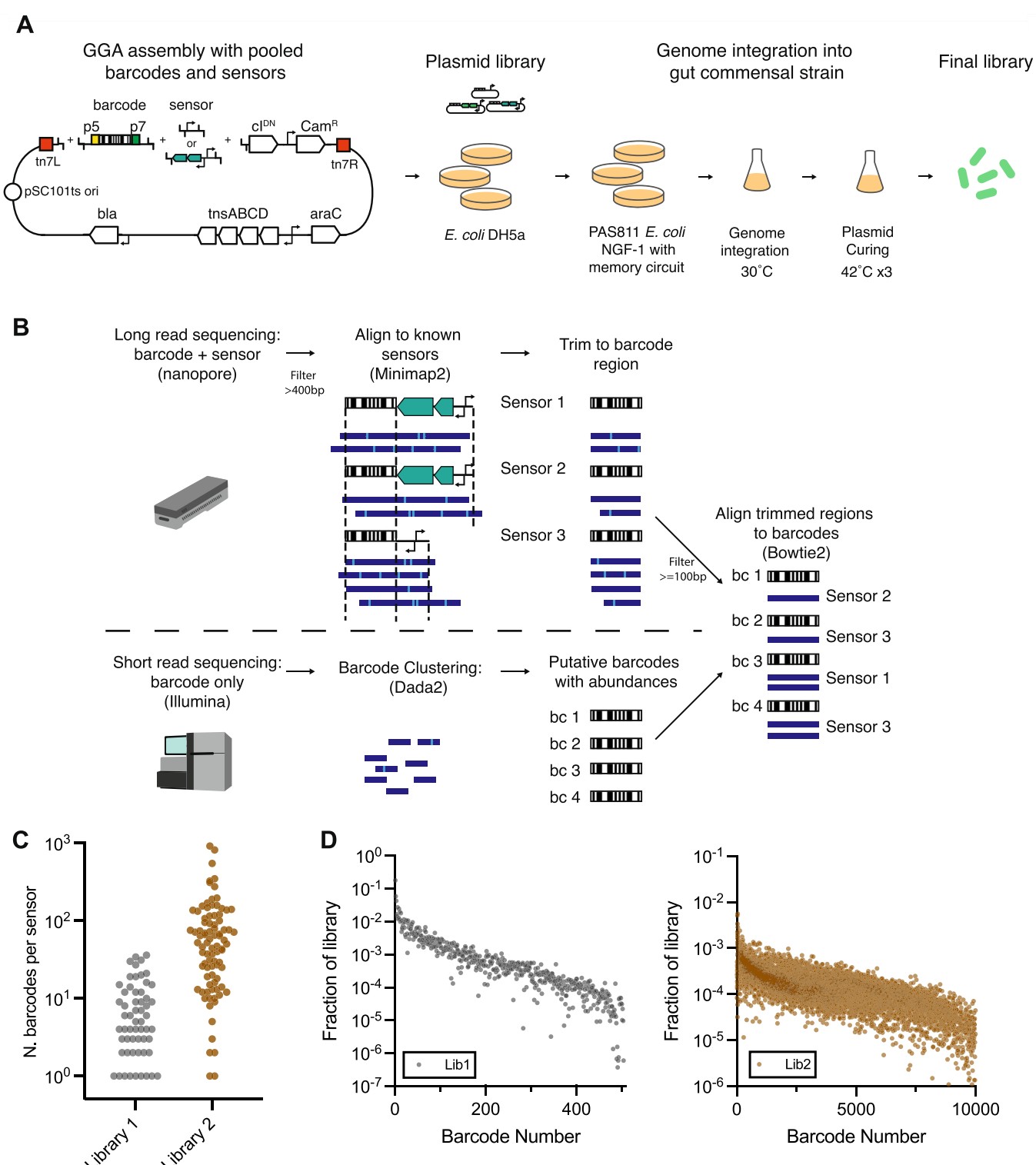

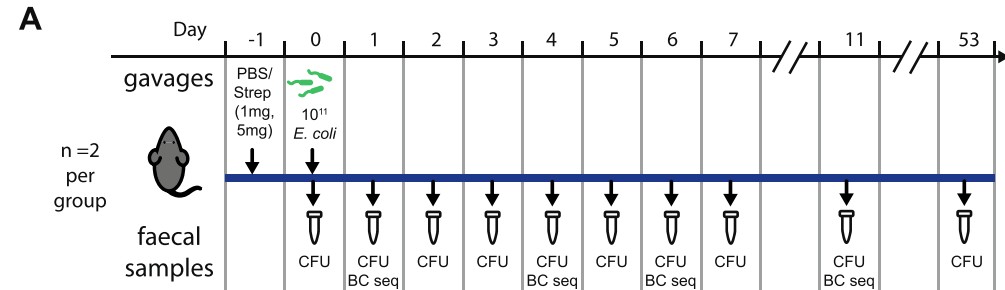

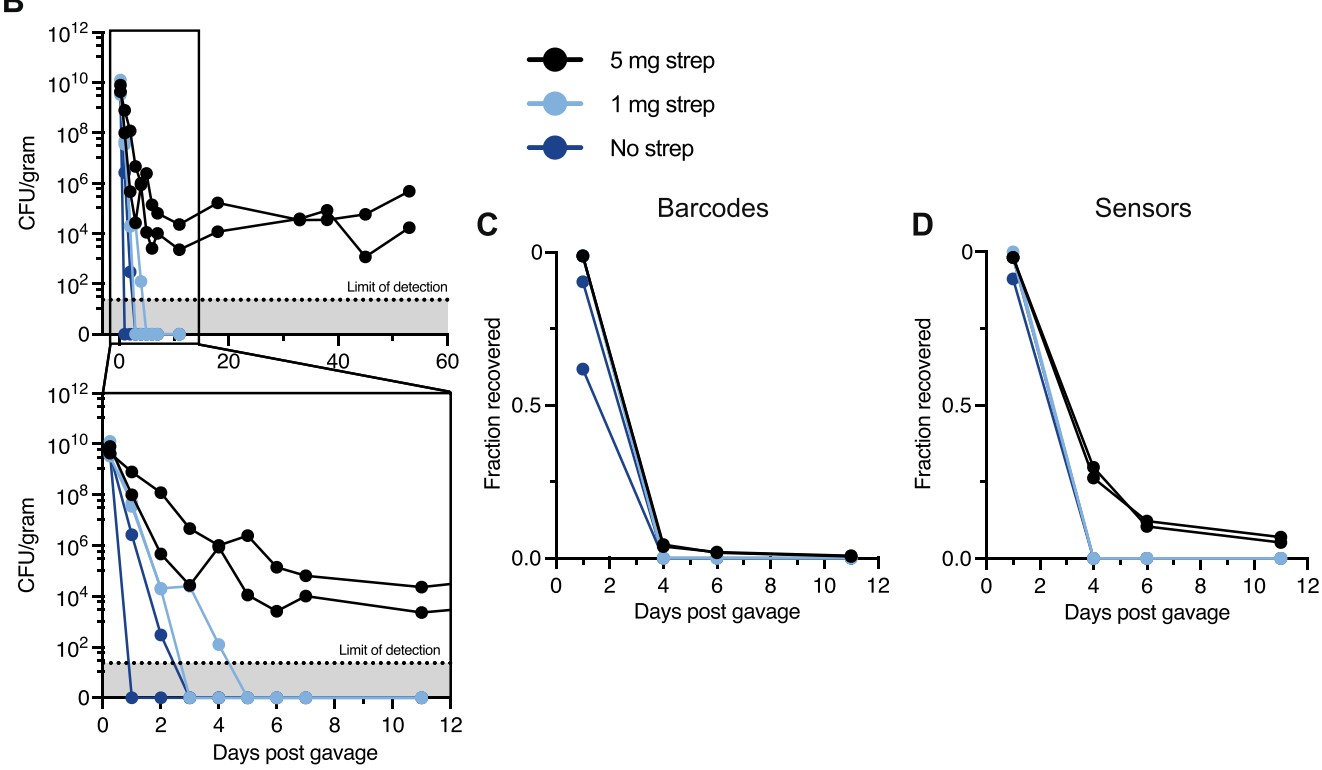

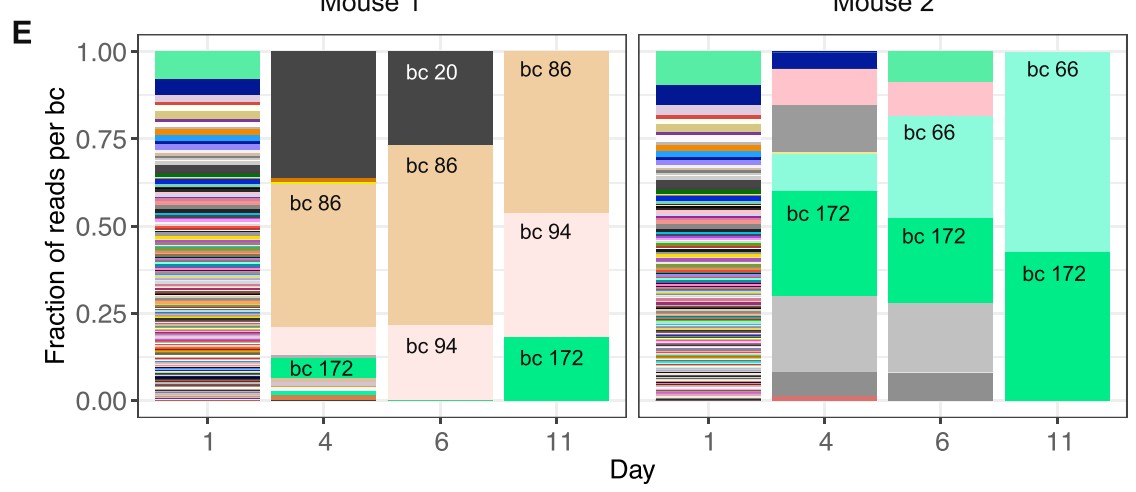

◀  **Figure EV2.  Barcode and strain diversity is maintained over short periods in the murine gut.**

(**A**) C57BL/6 mice ($n = 2$ per group) were administered different concentrations of streptomycin (0, 1 or 5 mg) via oral gavage, followed by ~$10^{11}$ Library 1 bacteria the following day. Faecal samples were measured for colonisation by plated CFU counts and for diversity by Illumina barcode sequencing. (**B**) CFU counts of engineered bacteria. (**C**) Fraction of total library barcodes and (**D**) fraction of total sensors recovered across the experiment. (**E**) Relative abundance of each barcode within two mice treated with 5 mg streptomycin. Individual barcodes with high abundances in later timepoints are numbered and highlighted for clarity.

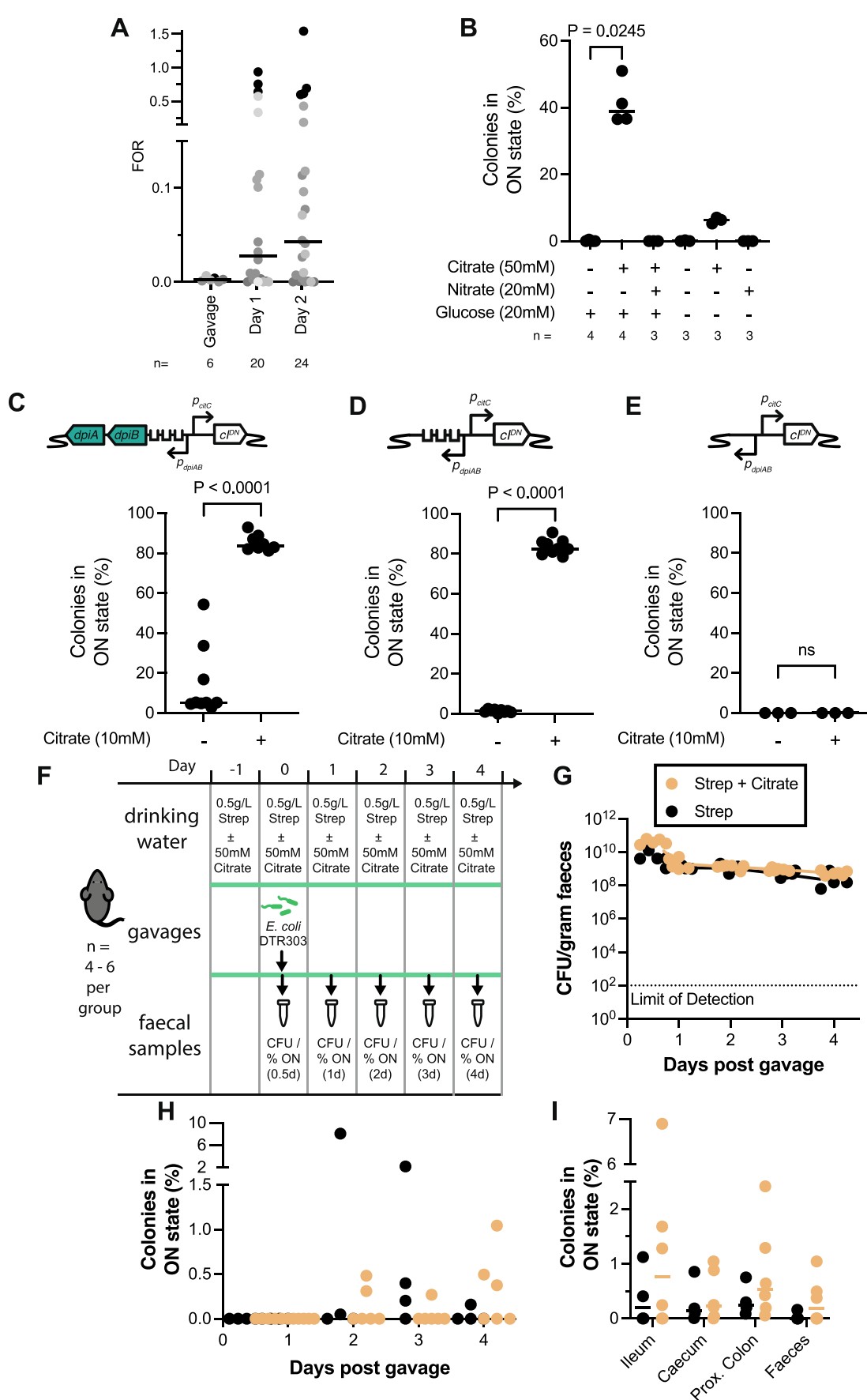

**Figure EV3.  Cr TCS2 biosensor response and validation in vitro and in vivo.**

(A) The response of Cr TCS2 (DpiAB -PcitC) sensors to the gut. Panels show FOR of individual barcodes ($n = 5$–8) in faecal pellets from each mouse ($n = 3$ D1 and $n = 4$ D2) & gavage ($n = 1$) sample passing QC. Shades correspond to unique barcode variants with total datapoints and median response marked. (B) Switching of DTR303, an individually cloned Cr TCS2 sensor, grown under in vitro anaerobic conditions in the presence of the regulator levels shown, as measured by indicator plating. Graph shows counts from $n = 3$–4 biological replicates (as labelled) with median marked. Statistics were calculated using a Kruskal-Wallis test with Dunn's multiple comparison correction comparing all groups to the glucose only control. (C) Indicator plating of Cr TCS2 sensor variants demonstrated similar response during anaerobic growth with 10 mM citrate between the full-length sensor DTR303 ($n = 9$) and (D) a truncation mutant DTR304 lacking the heterologous *C. rodentium* TCS genes dpiA and dpiB ($n = 9$). (E) Further truncation of DpiA-binding sites, DTR305, prevented switching ($n = 3$). Graphs in C-E show individual response values with median labelled. Statistics were calculated using a Mann–Whitney test, with significant P values shown. (F) DTR303 was administered to mice with streptomycin ($n = 4$) and streptomycin $+ 50$ mM citrate ($n = 6$) supplemented drinking water. (G) CFU counts of engineered bacteria (H) Colony response was measured on selective indicator plates from faecal pellets over 4 days following administration and (I) from dissected regions of the gut at endpoint. Graphs show median.

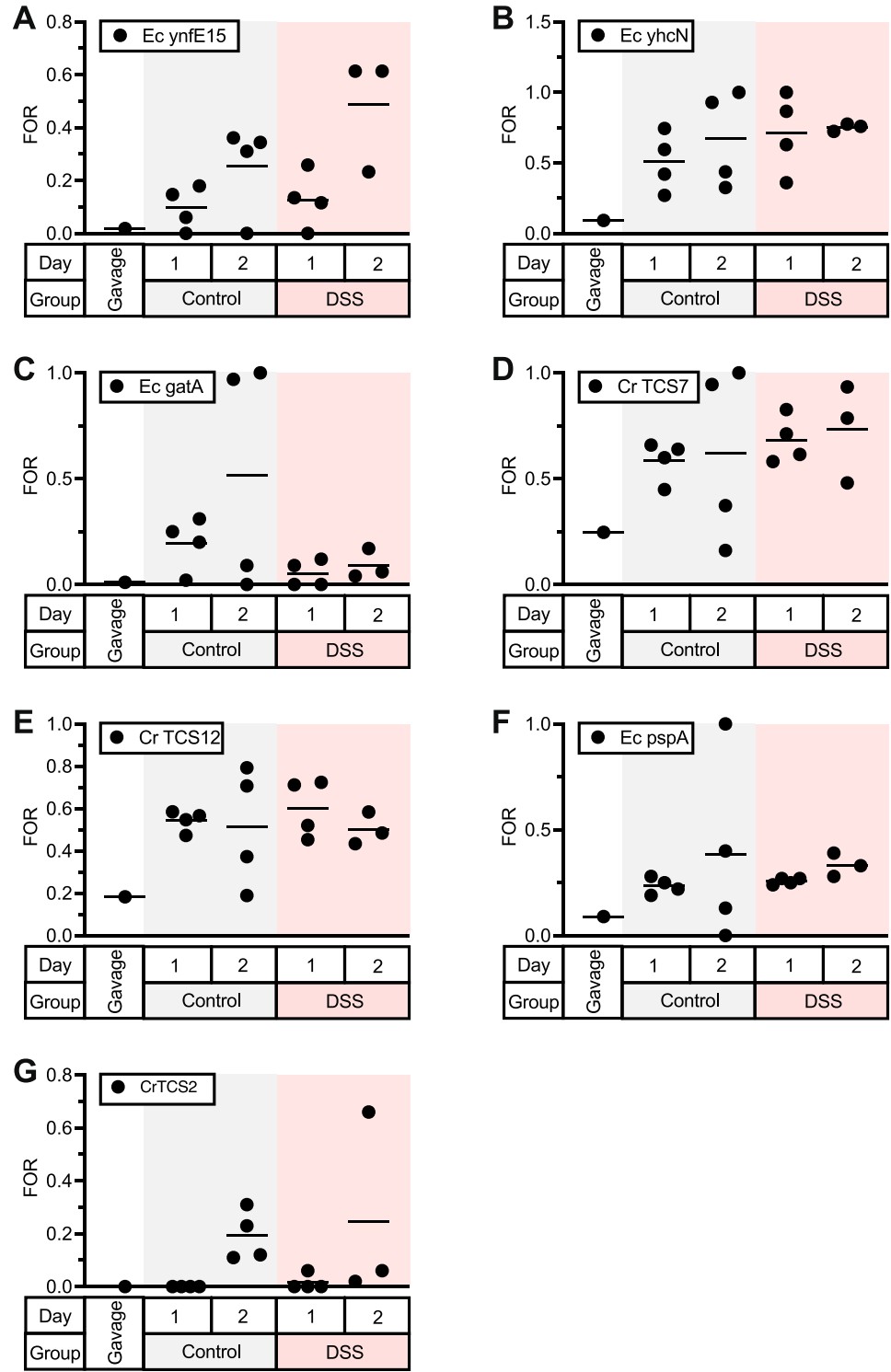

**Figure EV4. Screening of Library 1 + 2 identifies in vivo responsive biosensor candidates.**

Individual sensor activation on day 1 and 2 post bacterial administration (day 5 and 6 post DSS exposure) for (**A**) Ec ynfE15 control and top in vivo response sensors (**B**) Ec yhcN, (**C**) Ec gatA, (**D**) Cr TCS7, (**E**)) Cr TCS12, (**F**) Ec pspA and (**G**) Cr TCS. Graphs show individual mouse FOR values from datapoints passing sample and sensor QC criteria (for all sensors $n = 4$ for control and DSS day 1, $n = 3$ for DSS day 2 and $n = 1$ for gavage), with mean marked.

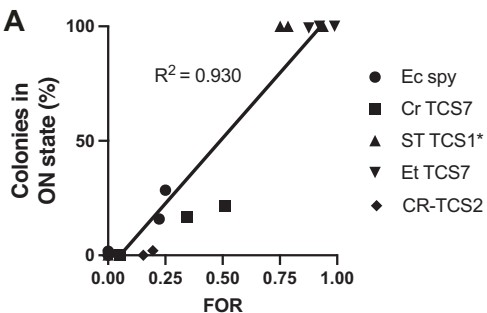

**Figure EV5.  Library screening results correlate with response of individually tested biosensors.**

(A) Comparison between FOR from pooled screening and colony counts of individual sensors tested under the same in vivo or in vitro conditions across all validated biosensors in this study. Graph shows mean FOR and mean response for 19 unique strain/condition combinations. A simple linear regression is overlaid.

