## [Peer Review File · Molecular Systems Biology]

A discovery platform for identification of host-induced bacterial biosensors from diverse sources

Clare Robinson, David Carreno, Tim Weber, Yangyumeng Chen, and David Riglar

Corresponding author(s): David Riglar (d.riglar@imperial.ac.uk)

Review Timeline:

Submission Date:	27th Aug 24
Editorial Decision:	1st Oct 24
Revision Received:	7th Mar 25
Editorial Decision:	10th Apr 25
Revision Received:	14th May 25
Accepted:	16th May 25

Editor: Poonam Bheda

Transaction Report:

1st Oct 2024

Manuscript Number: MSB-2024-12594

Title: A discovery platform accelerates identification of host-induced synthetic biosensors from diverse microbial sources

Dear Dr Riglar,

Thank you for the submission of your manuscript to Molecular Systems Biology. We have now received feedback from the three reviewers who agreed to evaluate your manuscript. As you will see from the reports below, the referees acknowledge the interest of the study and are overall supportive of your work; however they also comment on multiple aspects of the manuscript that should be strengthened in a revision.

Without repeating all the comments listed below, some of the more fundamental issues raised are the following:

- unclear benefit of the barcodes
- unclear advance over other methods
- unsure utility of the newly identified TCSs as biosensors

Importantly, during a cross-commenting session between the reviewers and the editor, a shared concern was that the current form of the manuscript is between a methods paper and a research paper, with too little information for a methods paper and too few new biological insights and validation to be a research paper. The consensus was that it would be beneficial to choose one angle and revise the text thoroughly as appropriate.

All other issues raised would need to be satisfactorily addressed. Please let me know in case you would like to discuss in further detail any of the comments, I would be happy to schedule a call.

We require:

1) A .docx formatted version of the manuscript text (including legends for main figures, EV figures and tables). Please make sure that the changes are highlighted to be clearly visible. Alternatively you may choose to submit your manuscript as a LaTeX file.

4) A .docx formatted letter INCLUDING the reviewers' reports and your detailed point-by-point responses to their comments. As part of the EMBO Press transparent editorial process, the point-by-point response is part of the Peer Review File (PRF), which will be published alongside your paper.

5) A complete author checklist, which you can download from our author guidelines (<https://www.embopress.org/page/journal/17574684/authorguide#submissionofrevisions>). Please insert information in the checklist that is also reflected in the manuscript. The completed author checklist will also be part of the PRF.

6) Please note that all corresponding authors are required to supply an ORCID ID for their name upon submission of a revised manuscript.

7) It is mandatory to include a 'Data Availability' section after the Materials and Methods. Before submitting your revision, primary datasets produced in this study need to be deposited in an appropriate public database, and the accession numbers and database listed under 'Data Availability'. Please remember to provide a reviewer password if the datasets are not yet public (see <https://www.embopress.org/page/journal/17574684/authorguide#dataavailability>).

This study includes no data deposited in external repositories.

8) All Materials and Methods need to be described in the main text using our 'Structured Methods' format, which is required for all research articles. According to this format, the Methods section includes a Reagents and Tools Table (listing key reagents, experimental models, software and relevant equipment and including their sources and relevant identifiers) followed by a

Methods and Protocols section describing the methods using a step-by-step protocol format. The aim is to facilitate adoption of the methodologies across labs. Please upload the Reagents and Tools table as a separate document when submitting your revised manuscript. More information on how to adhere to this format as well as a downloadable template (.docx) for the Reagents and Tools Table can be found in our author guidelines:

<https://www.embopress.org/page/journal/17444292/authorguide#structuredmethods>

9) For data quantification: please specify the name of the statistical test used to generate error bars and P values, the number (n) of independent experiments (specify technical or biological replicates) underlying each data point and the test used to calculate p-values in each figure legend. The figure legends should contain a basic description of n, P and the test applied. Graphs must include a description of the bars and the error bars (s.d., s.e.m.). Please provide exact p values.

10) Our journal encourages inclusion of *data citations in the reference list* to directly cite datasets that were re-used and obtained from public databases. Data citations in the article text are distinct from normal bibliographical citations and should directly link to the database records from which the data can be accessed. In the main text, data citations are formatted as follows: "Data ref: Smith et al, 2001" or "Data ref: NCBI Sequence Read Archive PRJNA342805, 2017". In the Reference list, data citations must be labeled with "[DATASET]". A data reference must provide the database name, accession number/identifiers and a resolvable link to the landing page from which the data can be accessed at the end of the reference. Further instructions are available at .

11) We replaced Supplementary Information with Expanded View (EV) Figures and Tables that are collapsible/expandable online. A maximum of 5 EV Figures can be typeset. EV Figures should be cited as 'Figure EV1, Figure EV2" etc... in the text and their respective legends should be included in the main text after the legends of regular figures.

<https://www.embopress.org/page/journal/17574684/authorguide#expandedview>

13) Author contributions: CRediT has replaced the traditional author contributions section because it offers a systematic machine readable author contributions format that allows for more effective research assessment. Please remove the Authors Contributions from the manuscript and use the free text boxes beneath each contributing author's name in our system to add specific details on the author's contribution. More information is available in our guide to authors.

14) Disclosure statement and competing interests: We updated our journal's competing interests policy in January 2022 and request authors to consider both actual and perceived competing interests. Please review the policy

<https://www.embopress.org/competing-interests> and update your competing interests if necessary.

Please also suggest a striking image or visual abstract to illustrate your article as a PNG file 550 px wide x 300-600 px high. Share synopsis text and image, as well as eTOC:

Please note that these would be the final versions and changes during proofing are usually not allowed

16) As part of the EMBO Publications transparent editorial process initiative (see our policy here:

https://www.embopress.org/transparent-process#Review_Process), Molecular Systems Biology will publish online a Peer Review File (PRF) to accompany accepted manuscripts.

In the event of acceptance, this file will be published in conjunction with your paper and will include the anonymous referee reports, your point-by-point response and all pertinent correspondence relating to the manuscript. Let us know whether you

agree with the publication of the PRF and as here, if you want to remove or not any figures from it prior to publication. Please note that the Authors checklist will be published at the end of the PRF.

Molecular Systems Biology has a "scooping protection" policy, whereby similar findings that are published by others during review or revision are not a criterion for rejection. Should you decide to submit a revised version, I do ask that you get in touch after three months if you have not completed it, to update us on the status.

I look forward to receiving your revised manuscript.

Yours sincerely,

Poonam Bheda, PhD
Scientific Editor
Molecular Systems Biology

Reviewer #1:

This paper focuses on identifying new biosensors that can sense and detect reporters of gut health. It involves the development of a new strategy for library construction that is based on Golden Gate cloning. The investigators proceed to screen their library for promoters that are induced under a variety of conditions, both in vitro and in vivo. The investigators primarily focus on identifying promoters activated in the gut in the absence and presence of inflammation. They also investigate whether they can identify promoters/TCSs induced in response to other cues, i.e., citrate, when administered to the gut. This approach does lead them to identify several new promoters, which recognize environmental cues with variable responses.

The conclusions drawn in this manuscript are convincing. The new approach to library cloning is clever and likely to be of interest to many. (Although it probably could be clearer if they included more general information on how Golden Gate cloning works). The investigators have identified several new TCS systems that can be further developed into biosensors. However, it is unclear whether they are responding at high enough levels to be used to develop sense and respond circuits. While this is not the paper's focus, the investigators state in the abstract that their studies are likely to provide information that can be used to design such sensors.

Major

1. The low oxygen control Ec ynfE15 sensor demonstrates only very weak activation in the absence of oxygen. Is this correct? How does one interpret a low FOR? While a statistically significant result, is it likely biologically relevant? How does the activity of Ec ynfE15 compare to the FNR promoter, which is often used to develop gut sense and respond circuits? In general, the discussion would benefit from some discussion regarding how one interprets the FOR related to different biosensors and how, in the future, others might use this information to develop sense and response circuits.
2. Lines 248-249: Does barcode variability reflect differences in the influence of the barcodes on the TCS responses vs noise in the TCS response? Given that the TCS has a very low response rate, perhaps it reflects noise in the circuit induction.

Minor

1. What is the rationale for labeling each TCS with multiple bar codes? Was this done because it enabled the construction of a library of barcoded TCSs in a pool as opposed to one at a time? Was it done to potentially deal with bottleneck issues? I think both ideas are discussed in the first paragraph of the discussion, which is great, although the ideas could be more clearly explained. For example, could point out how much more tedious it would be to make barcode strains one at a time.
2. Line 129: unless one is already familiar with Golden Gate cloning, they will not know the significance of the terms SAP or BsaI. Some additional background o
3. What are the X's in Fig. 2C representative of?
4. Line 414, this appears the first time the term GGA is used. And it appears to be the only time it is used outside of methods. Would just replace it with Golden Gate assembly.

Reviewer #3:

Summary:

In this manuscript, the authors develop a screening system for identifying potential active biosensors in the murine gut. They constructed a barcoded library of two-component system biosensors (TCSs) combined with the memory circuit of the lambda phage's lysis-lysogeny switch. The TCSs were computationally identified from diverse gut bacteria. They screen these libraries both in vitro and in vivo using mouse models to identify sensors of interest in response to different gut microbe inflammatory triggers. Through their methods, they identify and validate several inflammation-specific biosensors and conclude that their pipeline is effective at screening biosensors in the gut or other microbial environment.

Overall, this work provides value to the field by developing novel synthetic genetic memory circuits as biosensors within the mammalian gut. The authors present an interesting platform, but it is unclear what the direct impact and future application of their current work is to the broader field. There are some parts of the work that lacked clarity in why their chosen methods are significant improvements from other techniques were less clear to me (I apologize I missed that). Additionally, there are some parts of the work that lacked data and evidence that would have better supported their claims and conclusions. For example, longer-term mouse experiments were not performed on the identified biosensors from the 2-day in vivo screen. These features lessened my enthusiasm.

Major Questions/Comments:

- It is unclear from the manuscript the full scope and purpose of this work and why key choices were made. What is tricky about working with mammalian gut systems and why was this approach the best one for engineering biosensors in this area? Why were TCSs and memory circuits chosen for this work and how was the target molecule chosen? There is a lot of focus on what has worked in the past and what is easiest for construction of the libraries, but it is unclear why this is the best approach and what novelty it brings.
- The biosensors that the authors identified have known inflammatory triggers. The innovation of the biosensors is the circuit design, since what molecules regulate the TCSs that are used in the circuits is already known in literature. So, it is unclear why authors did not first test these libraries in vitro, identify winners, and validate in vivo given that their library size decreases in vivo due to population bottlenecks.

Minor Questions/Comments:

- Page 1 line 41: What hinders research in this area for gut biosensors?
- Page 2 line 67: What is considered a large library? Is 'hundreds' (line 28) considered large given many libraries for screening and selection are magnitudes larger.
- Page 2 line 73: Sentence is missing a reference.
- Page 2 lines 70-79: What is the advantage of using a TCS over other signaling pathways?
- Table 1: Why were some of the successfully amplified TCSs not categorized in either Library in this study?
- Page 5 line 140: there is a missing '1)' after included.
- Page 5 lines 158-9: If the end application of the biosensor is to sense tetrathionate in an anaerobic environment, why are both aerobic and anaerobic conditions tested? Additional clarification could strengthen the paper.
- Page 5 line 166: How diverse are the sizes between these biosensors?
- Figure 1A/B: The authors should consider adding more details and labeling because the cloning of memory circuits is unclear. For example, it is unclear how the "TCS" and "Inflammation associated promoters" in 1A are cloned together.
- Figure 2 C: What do the "X"s mean in the figure?
- Page 6 line 169: Only 1 specific biosensor was identified and validated, but the figure title uses plural "biosensors".
- Page 7 line 220: What limits the ability of the screen when they are constrained to 2-day experimental time frames?
- Figures 3 and 4: These figures are crowded and have many parts. The authors should consider putting some parts in the supplementary information or splitting the figure up into different figures.
- Page 8 Line 232: missing the word 'in' after "resulted".
- Figure 3E/F: The authors should perform a statistical significance test for these sensors for stronger support of sensor activation.
- Figure 3I: It is unclear from the figure caption what sensor is being measured here.
- Figure 4F: Statistical tests between control and DSS are missing.
- Page 13 line 421: If 65 different TCS were tested in total how were there hundreds of tested biosensors?
- Page 14 line 453: What is considered "potentially inaccessible environments"?
- Authors should double check citation formatting as it may be different from the standard format for Molecular Systems Biology.

Reviewer #4:

The manuscript reports a high throughput approach to screen for novel biosensors in the context of the mammalian gut. For this, Robinson and colleagues expanded on a previously developed biosensor screening platform based on a bacterial memory genetic circuit relying on the *cl/cro* promoters. The authors constructed libraries of heterologous two-component systems (TCSs) from various bacterial species identified by a custom computational pipeline. Unique DNA barcodes were incorporated to each cloned TCSs. Following in vitro and in vivo transcriptional screening, the authors reported several TCSs which showed activation in response to oxygen, tetrathionate, passing through a healthy or inflamed (DSS) mouse gut.

I found the general idea of the project exciting and of importance for the field. While the science appears technically sound, the paper is unfortunately rather difficult to read and understand. The observations made are quite simple, but they are presented, in my opinion, in a convoluted way and figures are sometimes unclear. This hinders understanding of the work presented and the main message of the study gets lost throughout the paper. Overall, it is my opinion that the manuscript in its current form does not present a convincing case that their system represents a significant advance compared to previous work. Title of the manuscript states that the discovery platform 'accelerates' identification of TCSs but I am not convinced this work does indeed accelerate, simplify or improve the process of TCSs characterization.

The major point is that the benefits of this approach over already available methods, including previous work from the senior author (Naydich et al., 2019), is not convincingly demonstrated. For instance,

- I assume this might be due to my own limitations in understanding the work presented here but after reading the full work I remain unsure of how the barcode system was actually useful. I understood that identical TCSs could be tagged with distinct barcodes, but I fail to grasp the reason why this could serve the purpose of the study. Sequencing can surely allow identification of the TCSs retrieved within the bacterial population and relative abundances of each TCSs can be determined. It is not clear how the barcodes were utilized to gain additional information. I am convinced the benefit of the barcodes exists, but I suggest the authors should rewrite the corresponding text to clarify this point. I can only assume that if this unclear for me, it might be as well for other readers.
- The sample Et TCS7 shown in figure 2 is demonstrated to respond to tetrathionate. While I appreciate that this TCS was not annotated, it is explained by the author that BLAST searches identified homology to a putative tetrathionate reductase promoter region (l. 192-193). This TCS could have then been quite easily identified *in silico* as a candidate to test for tetrathionate induction. Similarly, all identified TCSs in this study seem to have characterized homologs in other species, as shown in Fig. 2F, Fig. 3G, Fig. 3K and Fig. 4G.
- Unless I am mistaken, it appears in Fig. 4 that most of the biosensors detected to have a response to DSS or *in vivo* conditions are issued from the library 2, which are *E. coli*-derived sensors previously shown to respond to gut and gut inflammation related signals.

I am therefore doubtful of the benefit of using this complicated pipeline involving barcode sequencing etc. to identify these few TCSs which can be somewhat easily identified as candidates by BLAST search. I am not saying this method might not overall be useful, but I do not find the authors make presently a strong case for it.

Some minor comments regarding things that hindered my understanding of the work, in Figure 2:

- The color scale used in Fig. 2C is not appropriate. There is no way to distinguish FOR values between 4 and almost 0. Therefore, activation of the control *Ec ynfE15* for example, while clear on panel D, is virtually not visible at all on panel C and this figure is not as informative as it could be.
- My understanding is that panel D provides the same information as panel C for the *Ec ynfE15* sample, which is a positive control of known activity, so I would remove the former. It just makes the overall figure more complex than it needs to be and does not add any information for the reader.
- Similarly, I would remove panel E, which is just another way of representing the same data shown in panel C. The data of this panel is also not well plotted as legend indicates that medians are marked but they are not visible for the ST TCS1* samples which are in black color.
- Black crosses are shown in panel C but it is not explained what they mean. I would assume corresponding samples were not analyzed for some reasons, but it needs to be explained.
- I found the title of the figure too vague - 'identifies specific biosensors *in vitro*' does not convey much information. As far as I understand, the *in vitro* screening revealed mainly two biosensors of two types: Et TCS7 for tetrathionate and Cr TCS2 oxygen. I would rather be explicit in the title, indicating that these two promoters were identified via the screen. As it is currently worded, I found that the title hinders the comprehension of the data presented, which is already shown in a complicated way as I mentioned above.

Moreover, in Figure 3:

- Similarly to my previous comments, I would remove panel E and F, which are, I believe, redundant data from panel D and therefore not useful. This would simplify this figure that has too many panels.
- Panel I indicates 'control' as a group, while panel E and F show '5 mg strep'. Could the authors explain what they mean by 'control' here?
- It is not clear to me why switching *in vivo* of Cr TCS7 in DTR307 is consistent at 20% instead of higher values (l. 239-241). I think this should be discussed.
- It is not clear why sensors show a degree of barcode-to-barcode variability in their response (l. 248-250). This should be clarified.
- Authors should discuss Fig. 3O. It is not clear to me why only a small subset of sensor bacteria switch with addition to citrate (l. 269-271).
- Panel D shows only 3 mice for Day 1 but 4 at Day 2. This should be explained.
- Based on Fig. 2C, I would argue that the TCS Cr TCS2 appeared to be induced by anoxic conditions. However, the author do not mention this aspect in the corresponding section of the manuscript but only later discuss that this TCS appears to be induced in the murine gut (Fig. 3; l. 244-250). Authors then investigate extensively its regulation by mainly citrate but they do not discuss the relation with oxygen. I found this overall section of the manuscript confusing. It is not clear why this TCS was not

discussed simply as an oxygen sensor.

I have a few more specific minor comments/suggestions:

- Table 1 indicates a total of 163 'Grouped' TCSs while the text mentions 153 (l. 115).
- Numbering of additional trigger types starts at 2) while I assume the sentence should read "These included 1) E. coli promoter regions (...) 72 E. coli promoter regions of interest." (l. 140-145)
- It is not clear why TCSs with Bsal but not SapI cloning compatibility were added to Library 2 specifically (l. 140-142).

We thank the Reviewers for their constructive feedback. In response we have substantially modified and improved the quality and clarity of our manuscript. The text has been extensively reworked to better illustrate the challenges faced, the advantages of our approach in addressing these, and to provide clearer methodological details throughout the main text. We have also reworked all figures for clarity and included additional animal validation data and analysis to further strengthen the conclusions drawn. Below are our point-by-point responses.

Original reviewer comments are in black

Responses are in bold blue

Quoted text changes and corresponding lines are in blue

Reviewer #1:

This paper focuses on identifying new biosensors that can sense and detect reporters of gut health. It involves the development of a new strategy for library construction that is based on Golden Gate cloning. The investigators proceed to screen their library for promoters that are induced under a variety of conditions, both in vitro and in vivo. The investigators primarily focus on identifying promoters activated in the gut in the absence and presence of inflammation. They also investigate whether they can identify promoters/TCSs induced in response to other cues, i.e., citrate, when administered to the gut. This approach does lead them to identify several new promoters, which recognize environmental cues with variable responses.

The conclusions drawn in this manuscript are convincing. The new approach to library cloning is clever and likely to be of interest to many. (Although it probably could be clearer if they included more general information on how Golden Gate cloning works). The investigators have identified several new TCS systems that can be further developed into biosensors.

Thank you for highlighting these strengths of our work. We have now added additional detail in the text to more clearly introduce the golden gate cloning method.

Line 144: "Therefore, sensors were constructed by pooled golden gate assembly which uses Type IIS restriction enzymes, such as SapI and BsaI, that cut outside their recognition sites to afford ordered and scarless assembly of multiple DNA fragments in a single pot (Engler *et al*, 2008) (Figure EV1A)."

However, it is unclear whether they are responding at high enough levels to be used to develop sense and respond circuits. While this is not the paper's focus, the investigators state in the abstract that their studies are likely to provide information that can be used to design such sensors.

We agree with the reviewer that the degree of response of some sensors may impact their eventual application. However, a higher response within the faecal biosensor population doesn't inherently correspond to greater utility of that biosensor for all applications.

One example is the non-uniform nature of the gut environment. A key feature of 'sense-and-response' therapeutics is the potential to activate rapid and localised responses only at a site of disease, avoiding high-level and/or system wide exposure to therapies that may otherwise cause toxicity or off-target effects. Thus, a targeted but overall low percentage response may, in fact, be more favourable for these applications rather than less.

While outside the scope of this paper, where increased response is desirable, various options for downstream circuit optimisation could be implemented (eg. Tuning of ribosome binding site efficiency or addition of amplification modules). Our work seeks to alleviate the critical bottleneck of identifying promising inducible biosensors for further testing and optimisation.

We now point to some of these factors more clearly in the discussion:

Line 392: "Promising sensing components identified through HTMS-based screening can be optimised for this or other transcriptional outputs, including through dynamic tuning, which can be achieved for example through RBS modulation or the addition of genetic amplifier components (Wan *et al*, 2019)."

Major

1. The low oxygen control Ec ynfE15 sensor demonstrates only very weak activation in the absence of oxygen. Is this correct? How does one interpret a low FOR? While a statistically significant result, is it likely biologically relevant? How does the activity of Ec ynfE15 compare to the FNR promoter, which is often used to develop gut sense and respond circuits? In general, the discussion would benefit from some discussion regarding how one interprets the FOR related to different biosensors and how, in the future, others might use this information to develop sense and response circuits.

We have now added additional clarity to the interpretation of FOR to the text.

Line 188: "FOR ~1 corresponds to a fully ON sensor and ~ 0 to a fully OFF sensor, with intermediate values corresponding to sensors activated in at least a subset of the population. Incomplete activation may occur when exposure to inducers is limited in concentration or time, due to a promoter having low maximal expression, or from spatially variable regulation across the population such as may occur within the non-uniform gut. We have previously observed biologically meaningful activation of sensors in as low as ~1% of bacteria within a given population, so will consider FOR >0.01 as potentially activated in our screens."

In addition, using the combined data arising from this study we can confirm strong correlation between FOR values measured in screening and the

response of the same individually tested sensors during validation. We have add a discussion point, and corresponding figure speaking to this

Line 358: "The sensors tested across our study point to broad correlation between FOR levels measured during pooled screening and degree of response by an individually cloned sensor during validation (Figure EV5)."

In the revised manuscript we have simplified the *in vitro* screening section of the paper and no longer focus on *Ec ynfE15* response *in vitro*. With that said, we have previously demonstrated *in vitro* switching of a sensor based on this promoter in anaerobic growth in LB media (~12.5% of population) and cecal extract (Naydich et al. 2019. Msystems). We still use the sensor as a control for *in vivo* experiments, where FOR values are consistently higher than *in vitro*, mirroring results from our previous study (Naydich et al. 2019. Msystems).

It is unfortunately difficult to directly compare between activity of biosensors tested in a different genetic context, such as the PepT based biosensor that we believe the reviewer is referencing with regards to FNR regulation (Chien et al. 2021. Nat Biomed Eng).

2. Lines 248-249: Does barcode variability reflect differences in the influence of the barcodes on the TCS responses vs noise in the TCS response? Given that the TCS has a very low response rate, perhaps it reflects noise in the circuit induction.

It is exceptionally unlikely that a barcode would impact differential TCS response, nor any other strain specific response of a sensor. We do expect several potential sources of variability. Noise or stochasticity in gene expression across a population is certainly possible as suggested by the reviewer. Other potential sources include exposure of sub-populations to different microenvironments during transit of the non-uniform gut, circuit or genome mutations that could arise during cloning or growth of a given sensor isolate, and/or simple stochasticity in strain outgrowth of the bacteria.

We have added a point in the discussion reflecting this:

Line 349: "For some identified sensors, barcodes revealed heterogenous response between replicates. This variability may derive from exposure to different microenvironments during transit of the non-uniform gut, unexpected mutations in a given sensor isolate, stochasticity in gene expression, and/or variations in strain outgrowth of the bacteria. No matter the cause, identification of response variability is an important factor in prioritising sensors for further development and assessment."
"

Minor

1. What is the rationale for labeling each TCS with multiple bar codes? Was this done because it enabled the construction of a library of barcoded TCSs in a pool as opposed to one at a time? Was it done to potentially deal with bottleneck issues? I think both ideas are discussed in the first paragraph of the discussion, which is great,

although the ideas could be more clearly explained. For example, could point out how much more tedious it would be to make barcode strains one at a time.

The choice for labelling each TCS with multiple bar codes both allows for pooled library construction and the simultaneous assessment of multiple redundant sensor replicates (in effect technical replicates) within each experiment.

The choice to construct barcodes as a pooled golden gate aims to significantly reduce the cost and time for construction by allowing simultaneous barcoding in a single pooled reaction and with a single oligonucleotide. Combined consumable, reagent and staff time costs for cloning hundreds to thousands of individually designed barcoded strains would be entirely cost prohibitive for undertaking this study. We have added a statement to this effect:

Given the diverse array of potential, but hard to measure, sources for variability (as discussed above in Major point 2) averaging FOR values across >1 replicate is expected to increase the reproducibility of our screening results and thus reduce animal numbers required, among other benefits. An ability to assess variability between barcodes also provides valuable additional insight for sensor prioritisation. An example is seen in Figure 3F, where a single barcoded clone for Cr TCS7 was a clear outlier in screening. Unsurprisingly, our individually constructed clone used in validation reflected the behaviour of the other 6 barcoded isolates rather than the outlier.

We have considerably re-worked the manuscript to include more detail on the rationale for barcoding, including the use of multiple barcodes, across the introduction, results and discussion. These include statements such as:

Line 99: "The approach has several advantages, including.....; iii) increased confidence in sensor response data through construction of multiple clones with identical sensor sequence but unique barcodes for measurement in a single animal or test condition;

Line 142: "Construction of large libraries of individually cloned barcoded sensors is unfeasible due to cost and time constraints."

Line 345: "Cloning results in the simultaneous assembly of pooled sensors, each with multiple barcoded replicates. This facilitates assessment of sensor functionality and variability in each screen through measurement of replicate clones. The approach greatly reduces the cost and number of animals required for *in vivo* sensor testing and validation. For some identified sensors, barcodes revealed heterogenous response between replicates."

Regarding bottlenecking, having more barcodes for a given sensor will not, per se, increase the chances for a sensor to be retained in the population.

However, the resolution barcodes afford provides additional context and understanding to this phenomenon on a biological scale.

2. Line 129: unless one is already familiar with Golden Gate cloning, they will not know the significance of the terms SAP or Bsal. Some additional background o

As discussed above, we have now added additional detail in the text to more clearly introduce the golden gate cloning method.

Line 144: "Therefore, sensors were constructed by pooled golden gate assembly which uses Type IIS restriction enzymes, such as SapI and Bsal, that cut outside their recognition sites to afford ordered and scarless assembly of multiple DNA fragments in a single pot (Engler *et al*, 2008) (Figure EV1A)."

3. What are the X's in Fig. 2C representative of?

The X's represent insufficient sequencing data for that sensor/condition. The figure legends for relevant figures now includes the line:

"Squares marked X indicate insufficient sequencing read data for that sensor and condition."

4. Line 414, this appears the first time the term GGA is used. And it appears to be the only time it is used outside of methods. Would just replace it with Golden Gate assembly.

All references to golden gate assembly in the main text are now referred to in full, with the abbreviation GGA used solely in the methods.

Reviewer #3:

Summary:

In this manuscript, the authors develop a screening system for identifying potential active biosensors in the murine gut. They constructed a barcoded library of two-component system biosensors (TCSs) combined with the memory circuit of the lambda phage's lysis-lysogeny switch. The TCSs were computationally identified from diverse gut bacteria. They screen these libraries both in vitro and in vivo using mouse models to identify sensors of interest in response to different gut microbe inflammatory triggers. Through their methods, they identify and validate several inflammation-specific biosensors and conclude that their pipeline is effective at screening biosensors in the gut or other microbial environment.

Overall, this work provides value to the field by developing novel synthetic genetic memory circuits as biosensors within the mammalian gut. The authors present an interesting platform, but it is unclear what the direct impact and future application of

their current work is to the broader field. There are some parts of the work that lacked clarity in why their chosen methods are significant improvements from other techniques were less clear to me (I apologize I missed that). Additionally, there are some parts of the work that lacked data and evidence that would have better supported their claims and conclusions. For example, longer-term mouse experiments were not performed on the identified biosensors from the 2-day in vivo screen. These features lessened my enthusiasm.

Major Questions/Comments:

1. It is unclear from the manuscript the full scope and purpose of this work and why key choices were made. What is tricky about working with mammalian gut systems and why was this approach the best one for engineering biosensors in this area?

We have now made major revisions to the text to better explain the motivations and significance of the work. Regarding challenges working in the mammalian gut specifically, there are a broad set of challenges faced in the field. These include:

- 1) Measurement of potential metabolites to target as biomarkers is difficult due to spatiotemporal variations in metabolite concentrations throughout the gut, especially in faeces.**
- 2) Direct measurement of bacteria transcriptional response is complicated by lacking bacterial metatranscriptomic data and tools, especially in the presence of a full competing microbiome. The very short bacterial mRNA half-lives and the low abundance of bacterial mRNAs compared to host RNA and bacterial ribosomal RNA contribute here.**

We have included reference to these in the revised text:

Line 54: "Direct measurement of metabolic and transcriptional biomarkers in the gut is also challenging. The brevity of bacterial mRNA half-lives, which can average less than a minute (Jenniches *et al*, 2024), considerably limit the insights that can be gained from faecal bacterial transcriptional sequencing. Similarly, rapid spatiotemporal dynamics of the gut microbiota and microbiota-linked metabolites likely make the quantities of many metabolites in faeces unrepresentative of internal gut conditions (preprint:Carreño *et al*, 2024)."

Challenges relating to the use of engineered biosensors include:

- 3) A poor hit rate across the field for rationally designed sensors functioning as hoped *in vivo* without considerable optimisation. This includes a limited correlation between in vitro and in vivo sensor behaviour and is exacerbated by the limitations of knowledge on the gut and gut microbiota during disease.**

- 4) **Competition between engineered bacteria and native gut microbiota (many *in vivo* studies are tested in germ-free or gnotobiotic animals with drastically reduced microbiota)**
- 5) **Considerable costs associated with testing in animals, both financial and ethical. Thus, reducing the number of animals needed for identification and testing of a sensor is an important goal in and of itself.**

Additions to the text to better communicate these challenges include:

Line 47: “Sensors that have been prototyped and function well in controlled laboratory settings often perform poorly when exposed to the challenging growth and metabolic conditions of complex ‘real-world’ environments, such as the mammalian gut. Individual testing of new biosensors *in vivo* is also a resource intensive process, incurring high costs in the form of money, time, and number of animal’s used in research. Consequently, well-characterised biosensing circuits for the mammalian gut remain limited, hindering the complete fulfilment of this vision (Barra *et al*, 2020; Tanna *et al*, 2021).”

2. Why were TCSs and memory circuits chosen for this work and how was the target molecule chosen?

There is a lot of focus on what has worked in the past and what is easiest for construction of the libraries, but it is unclear why this is the best approach and what novelty it brings.

Of the many options for adding sensing capacity to engineered bacteria, which include TCSs, OCSs, native and synthetic promoter libraries, and engineered TFs, TCSs were chosen both as one of the most promising mechanisms for adding novel sensing functionality in a heterologous system and also as one of the most challenging sets of targets from a technical standpoint given their greater size diversity. By demonstrating the platform’s ability to accommodate diverse TCSs, we can infer the approach’s compatibility with other signalling components for future library development.

We have clarified this in the manuscript with the following statements:

Line 91: “TCSs are of particular interest as engineered sensing components because novel capacities can be transferred between bacterial species through simple introduction of TCS genes (Daeffler *et al*, 2017; Riglar *et al.*, 2017). Outside of *E. coli* grown in laboratory conditions, the vast majority of TCSs and their regulons remain poorly characterised, representing a large untapped pool of novel sensing capabilities. On a technical level, TCSs also provide a pool of sensor components with varying size that are challenging to track within a library context.”

Memory circuits were chosen based on their proven ability to address the limitations of faecal sampling, the challenges of spatiotemporal variation and the restricted metabolite and transcriptional sampling options for the

mammalian gut. These challenges are discussed more extensively in Major Question 1, above. This point is reflected in the introduction as:

Line 59: “Synthetic genetic memory circuits, ... are powerful tools for mitigating these challenges and providing non-invasive biosensor discovery, reporting and actuation.”

3. The biosensors that the authors identified have known inflammatory triggers. The innovation of the biosensors is the circuit design, since what molecules regulate the TCSs that are used in the circuits is already known in literature. So, it is unclear why authors did not first test these libraries *in vitro*, identify winners, and validate *in vivo* given that their library size decreases *in vivo* due to population bottlenecking.

While we agree with the reviewer than *in vitro* testing for response to specific metabolites is an important approach (eg. in Fig 2 we demonstrate this for tetrathionate), we disagree that this is inherently preferable for identifying *in vivo* responsive biosensors of interest. Indeed, the slow progress of “*in vitro* first” approaches is an inspiration for this study.

As discussed in Major Question 1, above, limited data exists to support rational metabolite or transcriptional biosensor selection for *in vivo* and disease responsive sensors. By example, while spy gene expression has been linked to inflammation in a previous study of mono-colonised gnotobiotic mice, several sensors that we screened from the same dataset with inflammation-specific induction, were some of our strongest hits as general responders to the healthy and inflamed gut. This could well be due to the background, complete microbiota in our tests. Irrespective of the cause, it is at best difficult, slow and costly to screen every putative hit, even *in vitro*.

Additionally, while strong data is available on regulatory pathways in type strains of *E. coli* grown in lab conditions, even in this case understanding of many regulators remains incomplete. The availability and reliability of this type of data rapidly decreases for promoters from other strains and species, or when *E. coli* is grown in diverse conditions such as the gut.

Our work aims to build a platform to expand our knowledge and available sensors (both finding new sensors and testing known sensors outside of the limited range within which they have been previously characterised). Instances where we have confirmed previously described or predicted interactions only serve to further validate the accuracy of the platform.

We have now reworked the introduction to better communicate these challenges and the motivation for our work, including statements such as :

Line 94: “Outside of *E. coli* grown in laboratory conditions, the vast majority of TCSs and their regulons remain poorly characterised, representing a large untapped pool of novel sensing capabilities”

Minor Questions/Comments:

1. Page 1 line 41: What hinders research in this area for gut biosensors?

The revised manuscript includes significant reworking of the introduction to better communicate the challenges our work addresses. We have elaborated on the statement in question to now include the following:

Line 46: “However, design, construction and testing of new bacterial biosensors is non-trivial. Sensors that have been prototyped and function well in controlled laboratory settings often perform poorly when exposed to the challenging growth and metabolic conditions of complex ‘real-world’ environments, such as the mammalian gut. Individual testing of new biosensors *in vivo* is also a resource intensive process, incurring high costs in the form of money, time, and number of animal’s used in research. Consequently, well-characterised biosensing circuits for the mammalian gut remain limited, hindering the complete fulfilment of this vision (Barra *et al*, 2020; Tanna *et al*, 2021).”

Additional challenges faced with regards to work in the mammalian gut more generally are discussed in Major Comment 1, above.

2. Page 2 line 67: What is considered a large library? Is 'hundreds' (line 28) considered large given many libraries for screening and selection are magnitudes larger.

Magnitude of library size is clearly context dependent. The section has been reworked during the revision for clarity. Given the qualitative nature of the statement, we have removed the word “large” in this instance, with the equivalent sentence now reading:

Line 78: “Memory ‘ON’ bacteria can thus be selected via growth in the presence of spectinomycin antibiotics or detected via β -galactosidase enzymatic activity.”

3. Page 2 line 73: Sentence is missing a reference.

References have now been added – the sentence is now Line 91.

4. Page 2 lines 70-79: What is the advantage of using a TCS over other signaling pathways?

The advantages of TCSs are discussed more fully in Major Question 2 above. The text now reads:

Line 91: “TCSs are of particular interest as engineered sensing components because novel capacities can be transferred between bacterial species through simple introduction of TCS genes (Daeffler *et al*, 2017; Riglar *et al.*, 2017). Outside of *E. coli* grown in laboratory conditions, the vast majority of TCSs and their regulons remain poorly characterised, representing a large untapped pool of novel sensing capabilities. On a technical level, TCSs also provide a pool of sensor components with varying size that are challenging to track within a library context.”

5. Table 1: Why were some of the successfully amplified TCSs not categorized in either Library in this study?

There are various potential reasons why an amplified sensor may not be identified in a library. It may have been successfully cloned but be present at too low abundance for barcode assignment, it may have been successfully cloned but toxicity/cellular burden prevented growth of that sensor and thus loss from the population, or it may not have been successfully cloned either due to random chance or a specific factor such as unfavourable DNA folding. We have included the following text in the discussion to further clarify this:

Line 400: "The remaining 20% may be absent based on stochastic abundance variations in the amplification and cloning process but could also result from toxicity or burden caused by the heterologous genes when introduced into *E. coli*, or differences in golden gate assembly efficiency during the pooled multiplexed library construction."

6. Page 5 line 140: there is a missing '1)' after included.

This has been corrected.

7. Page 5 lines 158-9: If the end application of the biosensor is to sense tetrathionate in an anaerobic environment, why are both aerobic and anaerobic conditions tested? Additional clarification could strengthen the paper.

From a validation perspective, we use both anaerobic growth in the absence of tetrathionate and aerobic growth in the presence of tetrathionate as negative controls for the ST TCS1* sensor. In the revision we have simplified this aspect of the study, focussing specifically on 0mM and 100m tetrathionate concentrations in the aerobic sample as a reflection of their greater importance as controls. This is reflected in the text as:

Line 200: "Mirroring our previous results, ST TCS1* sensors activated strongly and specifically when grown with tetrathionate in the absence of oxygen but not in the absence of tetrathionate or presence of oxygen (Figure 2A-B) (Riglar *et al.*, 2017)."

8. Page 5 line 166: How diverse are the sizes between these biosensors?

The range of sensor sizes has now been added to the text:

Line 342: "By incorporating a DNA barcoding strategy, using a single, hyper-variable oligonucleotide for barcode delivery, we generate easily sequenceable libraries of sensors of diverse sizes ranging from a few hundred to several thousand basepairs in a low-cost, single-pot reaction"

9. Figure 1A/B: The authors should consider adding more details and labeling because the cloning of memory circuits is unclear. For example, it is unclear how the "TCS" and "Inflammation associated promoters" in 1A are cloned together.

Figure 1 has been revised extensively to more clearly communicate the overall approach of the work. Figure EV1A has also been updated to more clearly describe the golden gate cloning procedure.

10. Figure 2 C: What do the "X"s mean in the figure?

The X's represent insufficient sequencing data for that sensor/condition. The figure legends for relevant figures now includes the line:

“Squares marked X indicate insufficient sequencing read data for that sensor and condition.”

11. Page 6 line 169: Only 1 specific biosensor was identified and validated, but the figure title uses plural "biosensors".

We disagree with this interpretation. While we focus on a single novel biosensor, the in vitro data tests the responses of all sensors under these conditions and identifies several sensors with varying response under the conditions tested: ST TCS1*, Et TCS7 and Ec ynfE15 included. It also validates the library approach from a methodological perspective. For clarity, we have modified the heading to now read:

“Barcoded biosensor screening of Library 1 identifies specific biosensor responses in vitro”

12. Page 7 line 220: What limits the ability of the screen when they are constrained to 2-day experimental time frames?

Data from our current and past work suggests that 2-days is a sufficient exposure period to induce sensing. The discussion mentions this as:

Line 427: “Nevertheless, the success of our approach suggests that library screening approaches are still feasible and valuable over short experimental periods (eg. up to 2 days). “

13. Figures 3 and 4: These figures are crowded and have many parts. The authors should consider putting some parts in the supplementary information or splitting the figure up into different figures.

The revised have been streamlined and simplified, with both Figures 3 and 4 being split into multiple figures. Figure 3 now covers mouse screening and CrTCS7 validation. Cr TCS2 (with revised mouse experiment data) has been moved to Figure EV3. Figure 4 includes in vivo screening only, with Figure 4 covering the Ec spy sensor validation.

14. Page 8 Line 232: missing the word 'in' after "resulted".

This sentence has been altered during editing of the revision.

15. Figure 3E/F: The authors should perform a statistical significance test for these sensors for stronger support of sensor activation.

The FOR results, which derive from library screening datasets, represent a challenge for robust and valid statistical testing. Not only do they represent multiple comparisons across the library of sensors and different conditions tested, but sensors also have differing numbers of barcodes assigned across the library.

To avoid over-interpretation, we have opted instead to focus on statistical testing only when validating sensor activity in individually cloned and tested sensors.

16. Figure 3I: It is unclear from the figure caption what sensor is being measured here.

The corresponding figure and figure caption (now Fig 3H) have been updated to clarify the panel refers to the DTR307 (Cr TCS7) sensor.

17. Figure 4F: Statistical tests between control and DSS are missing.

See above pt 15 for discussion of statistical testing of FOR values.

18. Page 13 line 421: If 65 different TCS were tested in total how were there hundreds of tested biosensors?

This statement refers to both TCS and promotor-based sensors across the libraries tested. We have clarified this in the text, which now reads:

Line 395: "The cloning strategy afforded libraries of hundreds (Library 1) or thousands (Library 2) of uniquely barcoded strains, tagging ~150 variant sensors, including TCS and promoter-based sensors"

19. Page 14 line 453: What is considered "potentially inaccessible environments"?

Due to the retention of memory, this approach is valuable for measuring any environments which cannot be constantly monitored. Specific examples have been added to the text, which now reads:

Line 434: "This pipeline not only enables the construction of biosensors tailored to the gut environment but also offers the potential for screening biosensors in other microbial growth permissible and potentially inaccessible environments, such as within microbial biofilms, tumours, or enclosed artificial systems such as bioreactors or microfluidic systems."

20. Authors should double check citation formatting as it may be different from the standard format for Molecular Systems Biology.

All aspects of the manuscript, including citations, have been revised based on MSB guidelines as per submission guidelines.

Reviewer #4:

The manuscript reports a high throughput approach to screen for novel biosensors in the context of the mammalian gut. For this, Robinson and colleagues expanded on a previously developed biosensor screening platform based on a bacterial memory genetic circuit relying on the *cl/cro* promoters. The authors constructed libraries of heterologous two-component systems (TCSs) from various bacterial species identified by a custom computational pipeline. Unique DNA barcodes were incorporated to each cloned TCSs. Following in vitro and in vivo transcriptional screening, the authors reported several TCSs which showed activation in response to oxygen, tetrathionate, passing through a healthy or inflamed (DSS) mouse gut.

I found the general idea of the project exciting and of importance for the field. While the science appears technically sound, the paper is unfortunately rather difficult to read and understand. The observations made are quite simple, but they are presented, in my opinion, in a convoluted way and figures are sometimes unclear. This hinders understanding of the work presented and the main message of the study gets lost throughout the paper. Overall, it is my opinion that the manuscript in its current form does not present a convincing case that their system represents a significant advance compared to previous work. Title of the manuscript states that the discovery platform 'accelerates' identification of TCSs but I am not convinced this work does indeed accelerate, simplify or improve the process of TCSs characterization.

The major point is that the benefits of this approach over already available methods, including previous work from the senior author (Naydich et al., 2019), is not convincingly demonstrated. For instance,

1. I assume this might be due to my own limitations in understanding the work presented here but after reading the full work I remain unsure of how the barcode system was actually useful. I understood that identical TCSs could be tagged with distinct barcodes, but I fail to grasp the reason why this could serve the purpose of the study. Sequencing can surely allow identification of the TCSs retrieved within the bacterial population and relative abundances of each TCSs can be determined. It is not clear how the barcodes were utilized to gain additional information. I am convinced the benefit of the barcodes exists, but I suggest the authors should rewrite the corresponding text to clarify this point. I can only assume that if this unclear for me, it might be as well for other readers.

Barcoding of sensors to allow identification of individual clones within the library using short-read sequencing alone has a range of benefits including cost of screening, reliability and robustness of data, and ease of use.

In terms of cost of screening and ease of analysis, barcoding allows for routine library tracking using more cost effective and higher fidelity Illumina sequencing. Given the length of TCSs (which can be thousands of base pairs), the diversity of sensor sizes within libraries (which can introduce bias in amplification if not the same length), and the potential for very subtle sequence variations between sensors (for example point mutations in ribosome binding sites to modulate translation rates) it is not practical nor reliable to amplify, sequence, and align sensor regions directly. Instead, barcoding allows for a single more intense barcode assignment process following which short read sequencing can be used for all subsequent experiments.

Regarding reliability and robustness of data, given the diverse sources for variability in sensor response (eg. exposure of sub-populations to different microenvironments during transit of the non-uniform gut, noise or stochasticity in gene expression, circuit or genome mutations that could arise during cloning or growth of a given sensor isolate, and/or simple stochasticity in strain outgrowth of the bacteria) averaging FOR values across replicates increases the robustness and reproducibility of our screening results. One benefit of this is to considerably reduce animal numbers required. An ability to assess variability between barcodes also provides valuable additional insight for sensor prioritisation. An example is seen in Figure 3F, where a single barcoded clone for Cr TCS7 was a clear outlier in screening. Unsurprisingly, our individually constructed clone used in validation reflected the behaviour of the other 6 barcoded isolates rather than the outlier.

These advantages are now more clearly communicated, for example:

Line 98: “DNA barcodes can be used to effectively identify clones within a library via short-read massive parallel sequencing. The approach has several advantages, including i) cost-effective, low-bias, simultaneous tracking of sensors, with broad flexibility for sensor size and genetic format; ii) the ability to reliably and accurately identify sensors with highly similar sequence composition, for example promoters with ribosome binding site (RBS) variations that may differ by only a few base pairs; iii) increased confidence in sensor response data through construction of multiple clones with identical sensor sequence but unique barcodes for measurement in a single animal or test condition; and iv) increased flexibility during testing for adding new sensors to existing libraries and/or testing combined libraries in a single experiment. “

2. The sample Et TCS7 shown in figure 2 is demonstrated to respond to tetrathionate. While I appreciate that this TCS was not annotated, it is explained by the author that BLAST searches identified homology to a putative tetrathionate reductase promoter region (l. 192-193). This TSC could have then been quite easily

identified in silico as a candidate to test for tetrathionate induction. Similarly, all identified TCSs in this study seem to have characterized homologs in other species, as shown in Fig. 2F, Fig. 3G, Fig. 3K and Fig. 4G.

Our goal of developing a screening platform focusses on using empirical testing to identify the sensors of interest worth detailed follow-up investigation and validation. Indeed, the activity of TCSs and inducible promoters of homologous genes are regularly found to vary, even between closely related species. Thus homology can inform us only to a certain degree.

The primary goal of the experiment in question was to validate the method through response of the control sensor ST TCS1* to tetrathionate in vitro. The identification of Et TCS7 as an un-annotated homologue that responds to tetrathionate in its native form with similar sensitivity to its strong-RBS optimised homologue serves primarily as a demonstration of the methods potential to identify unknown sensors, but also identifies a sensor of interest for further optimisation and investigation in its own right.

3 Unless I am mistaken, it appears in Fig. 4 that most of the biosensors detected to have a response to DSS or in vivo conditions are issued from the library 2, which are E. coli-derived sensors previously shown to respond to gut and gut inflammation related signals.

Response is combined with point 4 below.

4. I am therefore doubtful of the benefit of using this complicated pipeline involving barcode sequencing etc. to identify these few TCSs which can be somewhat easily identified as candidates by BLAST search. I am not saying this method might not overall be useful, but I do not find the authors make presently a strong case for it.

Of the 11 “top hits” we feature in Figure 4 for response in the gut or gut inflammation, 3 derive from our TCS pipeline (2 cloned in library 1, 1 cloned in library 2), 3 are RBS-optimised E. coli promoters from one of our previous studies (selected based on being promoters for genes essential to anaerobic respiration), 1 is the ST TCS1* from another of our previous studies, and 4 are E. coli promoters recently linked to gut inflammation by Randall Platt’s group (in gnotobiotic mice monocolonised with E. coli). The diversity of sources itself illustrates the flexibility for library combination afforded by our platform.

Of course, where possible, the sensors chosen for our libraries skew towards those most likely to respond under inflammatory conditions, given this is a primary goal for our study. Nevertheless, it’s important to note the limitations of available data from which sensors could be rationally designed. As in Point 2, above, the activity of TCSs and inducible promoters of homologous genes

are regularly found to vary, even between closely related species. Even where homologous gene induction is understood in relevant *in vivo* conditions this limits the insight gained from homology alone. As further discussed in Reviewer 3 - Major Question 1, above, few reliable *in vivo* metabolomic and transcriptomic datasets exist based on technical limitations of this complex environment.

By demonstration, of ~70 promoters in Library 2 sourced based on the same Platt-lab dataset, only 4 were identified in our top hits, and of these 3/4 responded strongly as general markers of the gut despite being identified as strong inflammation-specific markers in the previous study. This could be due to the extensive differences in the gut environment of monoclonised mice compared to the complex microbial environment of conventional mice tested in our study.

Given the considerable (we would argue infeasible) resources in terms of cost, time and animal usage it would take to individually clone and test such a large number of sensors in animal models in order to identify these hits, our data clearly point to the necessity and benefit of a screening pipeline for this purpose. By comparison, using our method we have managed to identify 11 top candidates from a pool of ~150 in a single 4 vs 4 screen within animals. Validation of two of these hits as specific and sensitive non-invasive biosensors of the gut and inflamed gut respectively in conventional animals demonstrates the power of this approach.

In the revised text we more explicitly discuss the source of promoters within each library, communicate the challenges we aim to address and the benefits of our screen, including:

Line 159: "In the first instance, barcodes were assigned for 49 unique SapI compatible TCSs. We supplemented these strains with 9 additional individually cloned 'curated' sensors identified in our previous studies (Naydich *et al.*, 2019; Riglar *et al.*, 2017) to afford 'Library 1' (Table 1, Data S1)."

Line 172: "we subsequently cloned Library 2, consisting of: 1) *E. coli* promoter regions previously induced in mouse models, including during inflammation in monoclonised mice (Schmidt *et al.*, 2022); 2) computationally identified TCSs that were not successfully assigned in Library 1 construction, including those with BsaI but not SapI cloning compatibility; and 3) several additional curated sensors of interest (Table 1, Data S1)."

Line 47: "Sensors that have been prototyped and function well in controlled laboratory settings often perform poorly when exposed to the challenging growth and

metabolic conditions of complex 'real-world' environments, such as the mammalian gut. Individual testing of new biosensors *in vivo* is also a resource intensive process, incurring high costs in the form of money, time, and number of animal's used in research."

Line 54: "Direct measurement of metabolic and transcriptional biomarkers in the gut is also challenging. The brevity of bacterial mRNA half-lives, which can average less than a minute (Jenniches *et al*, 2024), considerably limit the insights that can be gained from faecal bacterial transcriptional sequencing. Similarly, rapid spatiotemporal dynamics of the gut microbiota and microbiota-linked metabolites likely make the quantities of many metabolites in faeces unrepresentative of internal gut conditions (preprint:Carreño *et al*, 2024)."

Line 348: ". The approach greatly reduces the cost and number of animals required for *in vivo* sensor testing and validation"

Line 416: "Although many promoters were selected rationally based on prior evidence of upregulation or essentiality during inflammation, screening identified only a handful with promise as strong and specific sensors when used in this context within the conventional mouse gut."

Some minor comments regarding things that hindered my understanding of the work, in Figure 2:

- The color scale used in Fig. 2C is not appropriate. There is no way to distinguish FOR values between 4 and almost 0. Therefore, activation of the control Ec ynfE15 for example, while clear on panel D, is virtually not visible at all on panel C and this figure is not as informative as it could be.

This figure, and panel, has been revised to streamline communication of this aspect of the work. We now focus solely on the main control strain for this experiment, ST TCS1*. We present a reduced set of conditions and a revised scale of the heatmap which matches that used in subsequent figures.

- My understanding is that panel D provides the same information as panel C for the Ec ynfE15 sample, which is a positive control of known activity, so I would remove the former. It just makes the overall figure more complex than it needs to be and does not add any information for the reader.

This panel has been removed in the revised figure.

- Similarly, I would remove panel E, which is just another way of representing the same data shown in panel C. The data of this panel is also not well plotted as legend

indicates that medians are marked but they are not visible for the ST TCS1* samples which are in black color.

We disagree that this is a direct replicate of the data of panel C as it enables display of individual barcodes, which assist in understanding the variability of response between replicates. That information is not depictable in heatmap form and thus has value for display.

The graph has been revised to better display the median values.

- Black crosses are shown in panel C but it is not explained what they mean. I would assume corresponding samples were not analyzed for some reasons, but it needs to be explained.

The X's represent insufficient sequencing data for that sensor/condition. The figure legends for relevant figures now includes the line:

“Squares marked X indicate insufficient sequencing read data for that sensor and condition.”

- I found the title of the figure too vague -'identifies specific biosensors in vitro' does not convey much information. As far as I understand, the in vitro screening revealed mainly two biosensors of two types: Et TCS7 for tetrathionate and Cr TCS2 oxygen. I would rather be explicit in the title, indicating that these two promoters were identified via the screen. As it is currently worded, I found that the title hinders the comprehension of the data presented, which is already shown in a complicated way as I mentioned above.

The in vitro data tests the responses of all sensors and identifies several sensors with varying response under the conditions tested: ST TCS1*, Et TCS7 and Ec ynfE15 included. It also validates the library approach from a methodological perspective. To balance these purposes, we have modified the heading to now read:

“Barcoded biosensor screening of Library 1 identifies specific biosensor responses in vitro”

Moreover, in Figure 3:

- Similarly to my previous comments, I would remove panel E and F, which are, I believe, redundant data from panel D and therefore not useful. This would simplify this figure that has too many panels.

We have considerably modified and simplified Figure 3 during revision. As above, we have retained the barcoded replicate information contained in Panels E and F as it adds additional information not depicted in the simpler heatmap presentation.

To reduce figure complexity we have removed panels corresponding to Cr TCS2 follow up (which is now in Figure EV3).

- Panel I indicates 'control' as a group, while panel E and F show '5 mg strep'. Could the authors explain what they mean by 'control' here?

This label has been edited to 5mg strep.

- It is not clear to me why switching in vivo of Cr TCS7 in DTR307 is consistent at 20% instead of higher values (l. 239-241). I think this should be discussed.

There are various factors that may lead to less than 100% sensor response in vivo, including the non-uniformity of the gut environment (not all bacteria may be in the same region as the inducer), insufficient exposure to an inducer for complete memory formation across the population (eg. too low concentration or time), weak maximal induction by a given promoter preventing complete switching irrespective of exposure, or, in the case of promoters regulated by more than one TF, a combination of inducers and repressors may lead to intermediate expression.

As discussed more extensively in response to Reviewer 1, a higher response within the faecal biosensor population doesn't inherently correspond to greater utility of that biosensor for all applications.

These factors are now mentioned when first introducing the potential for partial switching:

Line 190: "Incomplete activation may occur when exposure to inducers is limited in concentration or time, due to a promoter having low maximal expression, or from spatially variable regulation across the population such as may occur within the non-uniform gut."

- It is not clear why sensors show a degree of barcode-to-barcode variability in their response (l. 248-250). This should be clarified.

As also discussed in point 1 above, various sources of variability between barcodes exist eg. exposure of sub-populations to different microenvironments during transit of the non-uniform gut, noise or stochasticity in gene expression, circuit or genome mutations that could arise during cloning or growth of a given sensor isolate, and/or simple stochasticity in strain outgrowth of the bacteria. This has been included in discussion

Line 349: "For some identified sensors, barcodes revealed heterogenous response between replicates. This variability may derive from exposure to different microenvironments during transit of the non-uniform gut, unexpected mutations in a

given sensor isolate, stochasticity in gene expression, and/or variations in strain outgrowth of the bacteria.”

- Authors should discuss Fig. 3O. It is not clear to me why only a small subset of sensor bacteria switch with addition to citrate (l. 269-271).

See answer below in combination with additional question on CrTCS2.

- Panel D shows only 3 mice for Day 1 but 4 at Day 2. This should be explained.

The Day 1 Mouse 4 sequencing did not pass our QC checks and so was removed from analysis. This is now indicated in the figure legend as:

“Sensor activation of top-ranked *in vivo* responsive sensors, and control sensors in mice on days 1 (n=3 passing QC) and 2 (n=4 passing QC) post library administration.”

- Based on Fig. 2C, I would argue that the TCS Cr TCS2 appeared to be induced by anoxic conditions. However, the author do not mention this aspect in the corresponding section of the manuscript but only later discuss that this TCS appears to be induced in the murine gut (Fig. 3; l. 244-250). Authors then investigate extensively its regulation by mainly citrate but they do not discuss the relation with oxygen. I found this overall section of the manuscript confusing. It is not clear why this TCS was not discussed simply as an oxygen sensor.

In the revised manuscript we have simplified the discussion and reduced the focus on Cr TCS2 by moving all validation data to Figure EV3. We have also replaced the Cr TCS2 *in vivo* testing experiment, which was previously tested in the absence of streptomycin, with a similar experiment undertaken to be more in line with the screening experiment by the presence of streptomycin. This has two benefits. The data now act as a more direct validation of the screening performed in Fig 3D. The higher colonisation levels also facilitate plating and counts of more colonies and thus more accurate quantification given the low switching numbers. The *in vivo* results validate the screening prediction, showing low levels of switching in the strep only control. Citrate supplementation does not measurably increase switching above strep alone.

Potential reasons for incomplete switching of sensors are discussed in the response to a point from the same Reviewer above. In the Cr TCS2 sensor’s case specifically, citrate uptake by the gut is expected to occur within the small intestine – potentially limiting exposure of the bacteria in other regions of the gut. Interestingly in some animals there appeared to be higher switching levels in samples from the Ileum and proximal colon compared to distal colon and feces, which is consistent with this expectation but ultimately

inconclusive given the low number of animals. There is also potential for oxygen and nitrate variability to regulate this sensor's response in vivo.

The revised text now discusses the response of this sensor to all conditions more wholistically and with reduced complexity.

Line 254: "Another biosensor of interest identified by screening was Cr TCS2, the *C. rodentium* *citC* promoter with divergent TCS DpiAB (Figure 3D). The 8 detectable Cr TCS2 linked barcodes showed a relatively low median response (<0.05 on both days), however, and high barcode-to-barcode variability in the mouse gut (Figure EV3A). *In vitro* screening also suggested some response under general anaerobic growth conditions (Figure 2A). The homologous TCS in *E. coli* is well characterised, with induction by citrate, anaerobiosis and glucose and inhibition by nitrate. Bacteria may be exposed to all these conditions within the gut.

When cloned and tested *in vitro* as an individual sensor (DTR303) we confirmed strong response to citrate in the presence of glucose and anaerobic conditions (Figure EV3B-C), strong inhibition by nitrate (Figure EV3B) and occasional switching in anaerobic conditions even without further induction (Figure EV3C). Truncation mutants further suggested that the *E. coli* host can regulate the heterologous promoter via DpiA (Figure EV3D-E). To validate the screening results and test whether dietary citrate supplementation could enhance induction, DTR303 bacteria were administered via oral gavage to mice (n=4-6 per group) fed streptomycin in their drinking water with or without the addition of 50mM sodium citrate (Figure EV3F). Colonisation levels were comparable between groups (Figure EV3G). Indicator plating suggested switching of a small subset of sensor bacteria in some mice from each group, as sampled from both faecal (Figure EV3H) and dissected gut contents (Figure EV3I). Citrate supplementation did not clearly increase biosensor response above that seen in streptomycin-only treated mice (Figure EV3H-I)."

I have a few more specific minor comments/suggestions:

- Table 1 indicates a total of 163 'Grouped' TCSs while the text mentions 153 (l. 115).

This error in the text has been amended to 163.

- Numbering of additional trigger types starts at 2) while I assume the sentence should read "These included 1) *E. coli* promoter regions (...) 72 *E. coli* promoter regions of interest." (l. 140-145)

This has been corrected in the revision.

- It is not clear why TCSs with Bsal but not SapI cloning compatibility were added to Library 2 specifically (l. 140-142).

Library 1 was constructed using SapI compatible triggers since the golden gate cloning process used the SapI restriction enzyme. At a later point we cloned additional TCSs which were not compatible with SapI (due to their sequences including SapI recognition sites) but were with BsaI using a different cloning vector and GGA overhangs. These were combined with a range of other sensors to form Library 2.

10th Apr 2025

Manuscript Number: MSB-2024-12594R

Title: A discovery platform for identification of host-induced bacterial biosensors from diverse sources

Dear Dr Riglar,

Thank you for the submission of your revised manuscript to Molecular Systems Biology. I am pleased to inform you that we will be able to accept your manuscript pending the following final amendments and appropriate response to reviewers:

- 1) In the main manuscript file, please include keywords to max. 5.
- 2) Please format the Data availability section according to the example below:
"The datasets and computer code produced in this study are available in the following databases:
- Chip-Seq data: Gene Expression Omnibus GSE46748 (<https://www.ncbi.nlm.nih.gov/geo/query/acc.cgi?acc=GSE46748>)
- Modeling computer scripts: GitHub (<https://github.com/SysBioChalmers/GECKO/releases/tag/v1.0>)
- [data type]: [full name of the resource] [accession number/identifier] ([doi or URL or identifiers.org/DATABASE:ACCESSION])"
- 3) Please rename the "Data and Materials Availability" section to "Data Availability"
- 4) Please rename "Competing Interests" to "Disclosure and competing interests statement". We updated our journal's competing interests policy in January 2022 and request authors to consider both actual and perceived competing interests. Please review the policy <https://www.embopress.org/competing-interests> and update your competing interests if necessary.
- 5) Our journal encourages inclusion of *data citations in the reference list* to directly cite datasets that were re-used and obtained from public databases. Data citations in the article text are distinct from normal bibliographical citations and should directly link to the database records from which the data can be accessed. In the main text, data citations are formatted as follows: "Data ref: Smith et al, 2001" or "Data ref: NCBI Sequence Read Archive PRJNA342805, 2017". In the Reference list, data citations must be labeled with "[DATASET]". A data reference must provide the database name, accession number/identifiers and a resolvable link to the landing page from which the data can be accessed at the end of the reference. Further instructions are available at https://www.embopress.org/competing-interests.
- 6) Please place individual sections of the manuscript in the following order: Title page - Abstract & Keywords - Introduction - Results - Discussion - Methods - Data Availability - Acknowledgements - Disclosure and Competing Interests Statement - References - Figure Legends - Expanded View Figure Legends.
- 7) For the figures and figure legends, please take care of the following:
 - Please note that figure 5 is mislabeled in the legends of the manuscript. This needs to be rectified.
 - Please note that the legend for figure EV3 H, I is missing in the manuscript. This needs to be rectified.
 - Please note that the exact p values are not provided in the legend of figure 5H (or in the figure itself)
 - Please note that information related to n is missing in the legends of figures 3E, F, H; 4D, F, G, H, I; EV3 A, EV4 A-G.
 - Please note that scale bar and its definition are missing for figure 2D.
- 8) Tables: Tables included in the main manuscript should be labeled as Table 1, 2, 3... Tables labeled as Table EVx should be uploaded as separate files.
- 9) Dataset EV legends: Each dataset will need its legend removed from the manuscript and added to the corresponding file in a separate tab.
- 10) Appendix file: Please add page numbers to the Table of Contents.
- 11) Appendix legends: The legends for Appendix figures should be removed from the main manuscript (they should only be present in the Appendix PDF below the corresponding figures).
- 12) Funding: Please note that funding information should be given in the "Acknowledgements" section (not in its own separate section).
- 13) Synopsis image: Please provide the synopsis image within the dimensions of 550 pixels wide x (300-600) pixels high. While we can resize it for you, when resized to 550 pixels wide, the image is not sufficiently high.
- 14) As part of the EMBO Publications transparent editorial process initiative (see our policy here: https://www.embopress.org/transparent-process#Review_Process), Molecular Systems Biology will publish online a Peer Review File (PRF) to accompany accepted manuscripts. This file will be published in conjunction with your paper and will include the anonymous referee reports, your point-by-point response and all pertinent correspondence relating to the manuscript. Let us know whether you agree with the publication of the PRF and as here, if you want to remove or not any figures from it prior to publication. Please note that the Authors checklist will be published at the end of the PRF.
- 15) After your paper is published, we will promote it on social media. If you have any handles or hashtags for Bluesky you would like included, please let us know.
- 16) Please provide a point-by-point letter INCLUDING my comments as well as the reviewer's reports and your detailed responses (as Word file).

I look forward to reading a new revised version of your manuscript as soon as possible.

Yours sincerely,

Poonam Bheda, PhD
Scientific Editor
Molecular Systems Biology

Reviewer #3:

The authors made a strong effort in the rebuttal and have better clarified their work with additional context and improved the data. They have addressed my major concerns.

Reviewer #4:

I appreciate the work the authors have clearly put into this revised version of their manuscript, which they have substantially improved by streamlining figures, providing details on the methods of their pipelines, clarifying the benefits of using the barcodes and overall simplifying the message of their manuscript by focusing on keys identified promoters. With this, it is my opinion that authors have appropriately addressed the concerns raised by the reviewer 1. They have also addressed most of my concerns; I remain however unsure of the novelty of their approach and its potential impact. While I do congratulate the authors on a technically impressive work, I would recommend that the authors perhaps slightly tone down the message of their manuscript and be more upfront about the potential caveats of their methods. Mainly:

- Out of the 11 top hits from library 1 shown in Fig. 4, only 3 were obtained from the TCSs pipeline, while the other 8 were sourced from previously published material where they were already suggested to be biosensor for the gut environment. It is therefore my understanding that about 3/4 of the promoters flagged as 'interesting' were actually not discovered by this pipeline. Similarly to my concern stated in the first round of review, I believe the approach explored in this manuscript might be overall promising, but I don't see that the authors make presently a strong case for it. It is my opinion that the authors should at least make it clear in the main text what percentage of the top hits discovered with their approach come from their pipeline (i.e. which ones are 'novel') and from what sources do these come from. The title of the manuscript indeed states that their approach is a discovery platform to identify bacterial biosensors from diverse sources, but it is unclear to me how much new bacterial biosensors were discovered here and if those came indeed from diverse bacterial sources.
- Additionally, in their discussion, lines 341-342, authors state that their work 'simultaneously evaluate thousands of uniquely barcoded biosensor strains'. While this is factually correct, I found the wording of this sentence potentially misleading. One could understand that the authors have tested thousands of different biosensors while my understanding is that 'only' about 65 distinct promoters were effectively tested in the end (as seen in their Table 1). The 'thousands' strains mentioned by the authors refers to various barcoded strains that correspond mainly to the same few promoters.

I also have some other minor specific comments:

- In Fig. 3, panels E, F and H, I recommend the 'group 5mg strep' line of the graphs to be removed. Here, the data displayed comes only from this one group of strep-treated mice, so this line does not add any value to the figure (in comparison to Fig. 4, where we compare data coming from control versus DSS group).
- In the legend of Fig. 5, I believe the authors did not update the numbering of their panels as the legend mentions panel I, J, K, L, M and N while it should refer to C, D, E, F, G and H instead.
- Similarly, legend of Fig. EV3 appears to not mention panels H or I and, unless I am mistaken, other mentioned panels appear incorrect (e.g. panel F is mentioned twice but refers to distinct data; one I assume refers to panel I?).
- Lines 159-160, could the authors explain what 'In the first instance, barcodes were assigned for 49 unique SapI compatible TCSs' mean?

Original reviewer comments are in black

Responses are in bold blue

Quoted text changes and corresponding lines are in blue

1) In the main manuscript file, please include keywords to max. 5.

Keywords added: Bacterial biosensor, gut microbiome, synthetic biology, inflammation.

2) Please format the Data availability section according to the example below:

"The datasets and computer code produced in this study are available in the following databases:

- Chip-Seq data: Gene Expression Omnibus GSE46748

(<https://www.ncbi.nlm.nih.gov/geo/query/acc.cgi?acc=GSE46748>)

- Modeling computer scripts: GitHub

(<https://github.com/SysBioChalmers/GECKO/releases/tag/v1.0>)

- [data type]: [full name of the resource] [accession number/identifier] ([doi or URL or identifiers.org/DATABASE:ACCESSION])"

This has now been updated.

3) Please rename the "Data and Materials Availability" section to "Data Availability"

This has now been updated.

4) Please rename "Competing Interests" to "Disclosure and competing interests statement". We updated our journal's competing interests policy in January 2022 and request authors to consider both actual and perceived competing interests. Please review the policy <https://www.embopress.org/competing-interests> and update your competing interests if necessary.

This has now been updated.

5) Our journal encourages inclusion of *data citations in the reference list* to directly cite datasets that were re-used and obtained from public databases. Data citations in the article text are distinct from normal bibliographical citations and should directly link to the database records from which the data can be accessed. In the main text, data citations are formatted as follows: "Data ref: Smith et al, 2001" or "Data ref: NCBI Sequence Read Archive PRJNA342805, 2017". In the Reference list, data citations must be labeled with "[DATASET]". A data reference must provide the database name, accession number/identifiers and a resolvable link to the landing page from which the data can be accessed at the end of the reference. Further instructions are available at

<https://www.embopress.org/page/journal/17574684/authorguide#referencesformat>.

All data citations are include in the suggested format (note: no change from previous version)

6) Please place individual sections of the manuscript in the following order: Title page - Abstract & Keywords - Introduction - Results - Discussion - Methods - Data Availability - Acknowledgements - Disclosure and Competing Interests Statement - References - Figure Legends - Expanded View Figure Legends.

This has now been updated.

7) For the figures and figure legends, please take care of the following:

- Please note that figure 5 is mislabeled in the legends of the manuscript. This needs to be rectified.

This has now been corrected

- Please note that the legend for figure EV3 H, I is missing in the manuscript. This needs to be rectified.

This has now been corrected

- Please note that the exact p values are not provided in the legend of figure 5H (or in the figure itself)

Exact p values are shown on the figure

- Please note that information related to n is missing in the legends of figures 3E, F, H; 4D, F, G, H, I; EV3 A, EV4 A-G.

Information on n has been added for all figures in legend and/or within the figure.

- Please note that scale bar and its definition are missing for figure 2D.

These have been added.

8) Tables: Tables included in the main manuscript should be labeled as Table 1, 2, 3... Tables labeled as Table EVx should be uploaded as separate files.

This has now been updated, and separate files for Table_EV1 and Table_EV2 provided.

9) Dataset EV legends: Each dataset will need its legend removed from the manuscript and added to the corresponding file in a separate tab.

This has now been updated

10) Appendix file: Please add page numbers to the Table of Contents.

This has now been updated

11) Appendix legends: The legends for Appendix figures should be removed from the main manuscript (they should only be present in the Appendix PDF below the corresponding figures).

This has now been updated

12) Funding: Please note that funding information should be given in the "Acknowledgements" section (not in its own separate section).

This has now been updated

13) Synopsis image: Please provide the synopsis image within the dimensions of 550 pixels wide x (300-600) pixels high. While we can resize it for you, when resized to 550 pixels wide, the image is not sufficiently high.

The dimensions of the image have been adjusted.

14) As part of the EMBO Publications transparent editorial process initiative (see our policy here: https://www.embopress.org/transparent-process#Review_Process), Molecular Systems Biology will publish online a Peer Review File (PRF) to accompany accepted manuscripts. This file will be published in conjunction with your paper and will include the anonymous referee reports, your point-by-point response and all pertinent correspondence relating to the manuscript. Let us know whether you agree with the publication of the PRF and as here, if you want to remove or not any figures from it prior to publication. Please note that the Authors checklist will be published at the end of the PRF.

We agree with the publication of the PRF and do not wish to remove any figures prior to publications.

15) After your paper is published, we will promote it on social media. If you have any handles or hashtags for Bluesky you would like included, please let us know.

@driglar.bsky.social

@clare-r.bsky.social

16) Please provide a point-by-point letter INCLUDING my comments as well as the reviewer's reports and your detailed responses (as Word file).

Provided here.

Reviewer #3:

The authors made a strong effort in the rebuttal and have better clarified their work with additional context and improved the data. They have addressed my major concerns.

We appreciate the reviewer's feedback.

Reviewer #4:

I appreciate the work the authors have clearly put into this revised version of their manuscript, which they have substantially improved by streamlining figures, providing details on the methods of their pipelines, clarifying the benefits of using the barcodes and overall simplifying the message of their manuscript by focusing on keys identified promoters. With this, it is my opinion that authors have appropriately

addressed the concerns raised by the reviewer 1. They have also addressed most of my concerns; I remain however unsure of the novelty of their approach and its potential impact. While I do congratulate the authors on a technically impressive work, I would recommend that the authors perhaps slightly tone down the message of their manuscript and be more upfront about the potential caveats of their methods. Mainly:

- Out of the 11 top hits from library 1 shown in Fig. 4, only 3 were obtained from the TCSs pipeline, while the other 8 were sourced from previously published material where they were already suggested to be biosensor for the gut environment. It is therefore my understanding that about 3/4 of the promoters flagged as 'interesting' were actually not discovered by this pipeline.

Similarly to my concern stated in the first round of review, I believe the approach explored in this manuscript might be overall promising, but I don't see that the authors make presently a strong case for it. It is my opinion that the authors should at least make it clear in the main text what percentage of the top hits discovered with their approach come from their pipeline (i.e. which ones are 'novel') and from what sources do these come from. The title of the manuscript indeed states that their approach is a discovery platform to identify bacterial biosensors from diverse sources, but it is unclear to me how much new bacterial biosensors were discovered here and if those came indeed from diverse bacterial sources.

While we appreciate the reviewer's concern regarding 'novelty' we believe that classifying the success of our method simply as a percentage of the top hits arising from one aspect of the study (the TCS identification pipeline) is an overly narrow view of the value of the work, as well as being a conceptually arbitrary statistic and as such of limited value to the reader.

For one, the 'top hit' threshold is not a clear, binary, distinction but a scale. It is possible that sensors that have not been highlighted in the main figures or text may still be of interest under certain circumstances. Similarly, we see no objective way to clearly define which sensors should be classed as "novel" from our study when varying degrees of information from an array of sources, strains and assays exists within the scientific literature.

As a library-based screening method, we instead focus on individually isolating/re-cloning and validating our sensors before making strong claims regarding their response. The successful validation of 3 sensors derived from our TCS pipeline (Et TCS7, Cr TCS7, Cr TCS2) and one from our E. coli promoter pool (Ec spy) under the same conditions as screened demonstrates the ability of our screen to identify true responders. That several additional sensors with logical mechanistic rationales for their response are identified during screen provides further confidence in these results. The fact that some of these sensors were designed (but in most cases not validated) in our previous studies does not diminish this but rather strengthens the evidence for their relevance given they have been re-cloned and tested under different conditions and in a new facility. It also directly demonstrates the benefits of our approach for recombination and mixing of sensors derived from different sources.

Additionally, there is an important distinction between a promoter found to be upregulated through transcriptomic measurements (in monoclonised mice), and a functional engineered bacterial biosensor strain as a tangible product.

To provide further clarity, however, we have added additional text in the discussion (line 375) to clearly state that torC, yeaR, and ST TCS1* sensors were based on designs first tested in our previous studies.

- Additionally, in their discussion, lines 341-342, authors state that their work 'simultaneously evaluate thousands of uniquely barcoded biosensor strains'. While this is factually correct, I found the wording of this sentence potentially misleading. One could understand that the authors have tested thousands of different biosensors while my understanding is that 'only' about 65 distinct promoters were effectively tested in the end (as seen in their Table 1). The 'thousands' strains mentioned by the authors refers to various barcoded strains that correspond mainly to the same few promoters.

We have now clarified at the beginning of our discussion that we've tested ~150 unique sensing components as a library consisting of thousands of uniquely barcoded biosensor strains.

I also have some other minor specific comments:

- In Fig. 3, panels E, F and H, I recommend the 'group 5mg strep' line of the graphs to be removed. Here, the data displayed comes only from this one group of strep-treated mice, so this line does not add any value to the figure (in comparison to Fig. 4, where we compare data coming from control versus DSS group).

The labelling of these figures along with the similar EV3A have been altered as suggested.

- In the legend of Fig. 5, I believe the authors did not update the numbering of their panels as the legend mentions panel I, J, K, L, M and N while it should refer to C, D, E, F, G and H instead.

This has now been updated

- Similarly, legend of Fig. EV3 appears to not mention panels H or I and, unless I am mistaken, other mentioned panels appear incorrect (e.g. panel F is mentioned twice but refers to distinct data; one I assume refers to panel I?).

This has now been updated

- Lines 159-160, could the authors explain what 'In the first instance, barcodes were assigned for 49 unique SapI compatible TCSs' mean?

Here we intended to specify that barcodes were assigned to 49 unique SapI compatible TCS before 9 additional individually cloned strains were added, and before the construction of Library 2. We have now removed "In the first instance" to help clearly communicate this.

16th May 2025

Manuscript number: MSB-2024-12594RR

Title: A discovery platform for identification of host-induced bacterial biosensors from diverse sources

Dear Dr Riglar,

Thank you again for sending us your revised manuscript. We are now satisfied with the modifications made and I am pleased to inform you that your paper has been accepted for publication.

Yours sincerely,

Sincerely,

Poonam Bheda, PhD
Scientific Editor
Molecular Systems Biology
